# Who Judges the Judge? LLM Jury-on-Demand: Building Trustworthy LLM Evaluation Systems

## Abstract

As Large Language Models (LLMs) become integrated into high-stakes domains, there is a growing need for evaluation methods that are both scalable for real-time deployment and reliable for critical decision-making. While human evaluation is reliable, it is slow and costly. Single LLM judges are biased, and static juries lack adaptability. To overcome these limitations, we propose LLM Jury-on-Demand - a dynamic, learning-based framework for scalable and context-aware evaluation. Our method trains a set of reliability predictors to assess when LLM judges will agree with human experts, leveraging token distributions, embeddings, and structural input features. This enables a fully adaptive evaluation where, for each data point, an optimal jury of the most reliable judges is dynamically selected and their scores are aggregated using their reliability as weights. Experiments on summarization and RAG benchmarks show that our dynamic jury system achieves significantly higher correlation with human judgment than both single-judge and static-jury baselines. These results highlight the promise of adaptive, learning-based juries for building scalable, more reliable and trustworthy evaluation systems for modern LLMs in high-stakes domains.[1]

## 1 Introduction

Large Language Models (LLMs) such as the GPT series, Llama, and Gemini have demonstrated transformative capabilities, leading to their rapid integration into critical, real-world applications (Brown et al., 2020; Touvron et al., 2023; Team et al., 2023). As these models are deployed in high-stakes domains, ensuring their outputs are reliable, safe, and aligned with human expectations has become a paramount concern (Shukla, 2025; Wang et al., 2023). The gold standard for assessing these qualities would be human evaluation, where experts provide nuanced judgments. However, this process is notoriously slow, expensive, and difficult to scale, making it impractical for the rapid development cycles of modern AI (Calderon et al., 2025). To overcome this scalability bottleneck, the field historically relied on reference-based automated metrics like BLEU and ROUGE, which measure lexical overlap between the generated output and a ground-truth reference text (Papineni et al., 2002; Lin, 2004). These methods are now widely considered insufficient for capturing multifaceted attributes like completeness, relevance, or groundedness in the sophisticated outputs of modern generative models (Zhang et al., 2019; Cao et al., 2025).

To address this evaluation gap, researchers have increasingly adopted the LLM-as-a-Judge paradigm, which leverages powerful language models like GPT-4 to serve as scalable, automated evaluators (Zheng et al., 2023; Li et al., 2024b; Gu et al., 2024). While promising, this approach introduces a critical trade-off where the scalability of a single LLM judge comes at the cost of reliability. The papers Schroeder & Wood-Doughty (2024), Li et al. (2024a) and Baumann et al. (2025) contain substantial evidence showing that single judges can be prone to systematic biases and inconsistencies, limiting their trustworthiness. A logical evolution has been to employ a "jury" of multiple LLMs to improve robustness (Feng et al.,

---

[1]The views expressed in this paper are solely those of the authors and do not necessarily reflect the views of their affiliated institutions.

2025; Verga et al., 2024). However, these jury systems typically rely on static aggregation methods, such as simple averaging. This fails to address a more fundamental issue, as a judge's expertise is context-dependent and its reliability can change dramatically based on the text being evaluated. This leaves a critical gap for a truly adaptive evaluation system.

In this paper, we introduce LLM Jury-on-Demand, a novel framework that bridges this gap by creating a dynamic, learning-based evaluation system. Our work moves beyond static juries by training a system to predict the reliability of each potential judge based on a rich set of features extracted from the text. This allows our framework to perform a fully adaptive evaluation where, for each data point, an optimal jury of the most reliable judges is dynamically selected, and their scores are aggregated using their reliability as weights. Our main contributions are threefold:

- A new framework for adaptive LLM evaluation that demonstrates superior correlation with human judgment compared to single-judge and static-jury baselines.
- A method to predict LLM judge reliability at the instance level using text-based features.
- Extensive experiments and analyses across multiple tasks and datasets to validate the effectiveness of the proposed approach.

## 2 RELATED WORK

The evaluation of large language models is a rapidly evolving field, as captured in recent surveys mapping the transition from static benchmarks to more dynamic and automated evaluation frameworks (Cao et al., 2025). Our work builds upon three key research areas: the LLM-as-a-Judge paradigm, the evolution from single judges to multi-model juries, and the broader concept of LLM performance prediction.

The LLM-as-a-Judge approach has become a scalable alternative to human annotation (Zheng et al., 2023), with surveys documenting its widespread application and promising correlation with human preferences(Li et al., 2024b; Gu et al., 2024). However, this paradigm has significant limitations. LLM judges exhibit biases, such as a preference for longer answers and sensitivity to the order in which responses are presented (Schroeder & Wood-Doughty, 2024), and their judgments can be skewed by their own intrinsic style or pre-training data, which compromises the fairness and reliability of the evaluation (Li et al., 2024a). These challenges motivate the need for more robust frameworks that can mitigate the inherent biases of a single judge.

To address these limitations, a growing body of work has explored using a "jury" of multiple LLMs, based on the insight that collaboration among diverse models can lead to more stable and reliable assessments (Feng et al., 2025). Initial work shows that simple ensembles, such as averaging the scores from a panel of smaller models, can outperform a single, larger model at a lower cost(Verga et al., 2024; Rahmani et al., 2024). More advanced methods have explored multi-agent frameworks where judges engage in peer-review or debate-like discussions to arrive at a consensus (Chu et al., 2024; Zhao et al., 2024). While a significant step forward, they typically rely on either static aggregation methods like simple voting or averaging or require complex and often unscalable conversational interactions. They do not account for the fact that a judge's expertise varies across different contexts, leaving a critical gap for systems that can adapt the jury's composition and weight to the specific context of the text being evaluated.

Our work is also grounded in LLM performance prediction. Studies have shown that it is possible to train a model to predict an LLM's performance on a given task by using features derived from the model and the task itself (Ye et al., 2023). Some approaches have even trained "assessor" models to predict when another model is likely to answer a question correctly, a concept that parallels our goal of predicting reliability (Schellaert et al., 2025). While these works validate the fundamental premise that LLM performance has learnable patterns, they typically focus on predicting general, task-level success rather than the instance-level reliability of an LLM acting as an evaluator. Our framework innovates by

applying this concept to jury-based evaluation, enabling the dynamic selection and weighting of judges on a per-instance basis.

## 3 METHODOLOGY

In this section, we detail the architecture and components of LLM Jury-on-Demand framework. Our framework is designed to produce more reliable automated evaluations by shifting from a static to a dynamic, learning-based process. The central hypothesis of our work is that an LLM judge's reliability is not fixed, but varies based on the specific characteristics of the text it evaluates. Our system models this variance by learning to predict when each judge is likely to agree with human experts.

### 3.1 FRAMEWORK OVERVIEW

The LLM Jury-on-Demand framework operates through a multi-stage pipeline, as shown in Fig. 1. The process begins with a set of input texts related to the evaluation task. Crucially, the set of texts we analyze depends on the task itself. For summarization tasks, the system analyzes both the original source text and the model-generated output summary. For Retrieval-Augmented Generation (RAG) tasks, it analyzes the source text, the retrieved context, and the final generated answer. This context-rich approach allows the system to capture a more complete picture of the evaluation challenge.

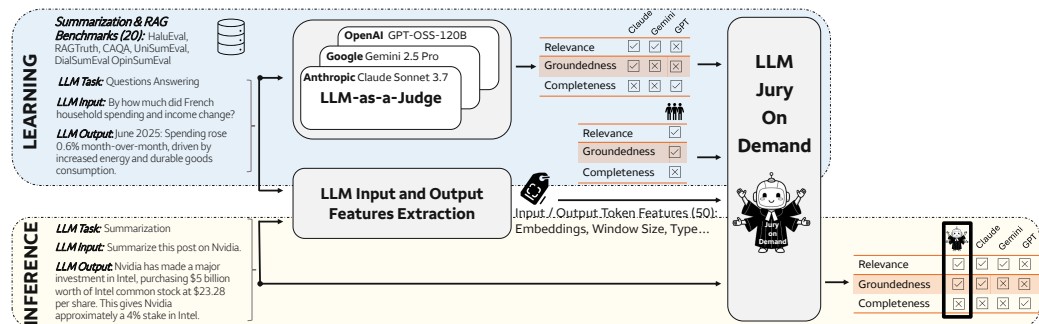

Figure 1: Overview of the LLM Jury-on-Demand inference pipeline. The system extracts features from input texts to predict judge reliability, dynamically assembles a jury of the top $K$ most reliable judges for each instance, and calculates a final weighted score.

These texts are first processed by a feature extraction module, which extracts a wide range of textual and semantic signals. The resulting feature vector for each instance serves as input to a suite of pre-trained reliability prediction models. In the final stage, the system leverages these reliability predictions to perform a fully adaptive, per-instance evaluation. For each data point, a jury of a pre-tuned size $K$ is dynamically assembled by selecting the judges with the highest predicted reliability for that specific instance. The final score is then computed as a weighted average of these selected judges' raw scores, using their reliability predictions as the weights.

### 3.2 FEATURE ENGINEERING FOR RELIABILITY PREDICTION

The foundation of our system is its ability to represent the evaluation context through a rich set of predictive features. We hypothesize that signals related to a text's size, complexity, and semantic content can reveal the scenarios in which different LLM judges excel or struggle. The features are extracted from all available texts (source, context, and output, as applicable to the task) and concatenated into a single feature vector for each data point. Many of these features are computed using Natural Language Toolkit (NLTK) (Bird et al., 2009). A complete list of features is in the Appendix A. Key feature categories are:

**Text Size Features:** These include basic structural metrics such as word count, sentence count, paragraph count, and the compression ratio between the source and generated text.

**Special Words Features:** These count the occurrences of specific word types that can indicate the style or complexity of the text. Examples include counts of difficult words (words that have more than two syllables), named entities and modality verbs (e.g. "could", "should") etc.

**Text Complexity Features:** These quantify readability and ambiguity using established linguistic formulas. Examples include the Flesch reading ease index (Kincaid et al., 1975), lexical diversity (the variety of words used), and other measures of syntactic and semantic ambiguity.

**Embedding-Related Features:** Embeddings encode text into a dense vector representation (Mikolov et al., 2013; Pennington et al., 2014; Devlin et al., 2019). These capture the semantic meaning and topic of the text. Top 10 PCA components (Jolliffe, 2011) of text embeddings are used as features. Additionally, we compute cosine similarity between each text component's embedding and a set of predefined topic embeddings (e.g. finance, technology), using these similarity scores as additional topical relevance features.

### 3.3 LEARNING TO PREDICT JUDGE RELIABILITY

Our framework learns judge reliability by training a dedicated machine learning model for each specific evaluation context. That is, for each potential judge, for each task (e.g., summarization), and for each evaluation metric (e.g., completeness), we train a distinct model. The purpose of this model is not to predict the evaluation score itself, but rather to predict the probability that the corresponding judge will be reliable on that metric for a given data point.

We frame this as a binary classification task to predict whether a judge's score will be "good" or "bad". To generate the ground-truth labels for training, we compare each judge's score against a gold-standard human expert score. First, we apply min-max normalization to all human and model scores to scale them to a $[0, 1]$ range. A judge's evaluation is then labeled as "good" (1) if its normalized score falls within a predefined tolerance hyperparameter, $\tau$, of the normalized human score. Otherwise, it is labeled as "bad" (0).

For each classification model, we use XGBoost, a gradient-boosted tree algorithm known for its strong performance on tabular data (Chen & Guestrin, 2016). This approach is grounded in the broader research area of LLM performance prediction, which has shown that model performance can be learned from features (Ye et al., 2023; Schellaert et al., 2025). At inference time, our trained models output a probability score between 0 and 1, which we use as the predicted reliability.

### 3.4 ASSEMBLING AND SCORING THE JURY

The core of our framework is its ability to assemble an expert jury and use its members' dynamically weighted opinions to compute a final score. This process, which is applied for each individual data point, involves two key mechanics: a reliability-based jury selection algorithm and an instance-level dynamic weighting scheme.

**Jury Selection.** For each data point, we first use the suite of pre-trained reliability models (described in Sec. 3.3) to generate a reliability score for each of the $N$ judges in our pool. A jury of a pre-tuned size $K$ is then formed by simply selecting the $K$ judges with the highest predicted reliability scores for that specific instance. This approach allows the jury's composition to be completely dynamic, adapting to the unique characteristics of each text.

**Dynamic Score Aggregation.** Once the instance-specific jury is selected, the final evaluation score is calculated as a weighted average of the raw scores from those $K$ jury members. The weights are derived directly from the instance-specific reliability scores, $[r_1, r_2 \ldots, r_K]$. Specifically, the weight for judge $i$ in the jury is calculated as $w_i = r_i/(\sum_{j=1}^{K} r_j)$. This dynamic, per-instance selection and weighting process allows our system to prioritize the

opinions of the most trustworthy judges for any given text, which stands in contrast to prior systems that rely on static juries and static aggregation methods like simple averaging (Verga et al., 2024).

These two mechanics are the building blocks for our system's training and inference pipelines.

### 3.5 System Training and Inference

Our framework involves a one-time training and tuning phase to establish an optimal configuration, which is then used in a repeatable inference pipeline to evaluate new data points.

**Training and Hyperparameter Tuning.** The goal of the training phase is to find the optimal hyperparameters that will generalize best. This involves finding the single best jury size $K$ and the optimal tolerance level $\tau$ for each individual judge's reliability model. To do this, we first train a large pool of reliability predictor models on our training data, covering all possible combinations of judges and tolerance values. We then conduct a search over the hyperparameter space of possible jury sizes ($K$) and per-judge tolerance configurations. Each configuration is evaluated on a held-out validation set. For every data point in the validation set, we apply the *Jury Selection* and *Dynamic Score Aggregation* mechanics described in Sec. 3.4. The configuration that yields the highest Kendall's Tau correlation with human scores across the validation set is selected as the optimal configuration for the final system.

**Inference Pipeline.** With the optimal configuration locked in (i.e., a fixed $K$ and a set of optimal reliability models), the system is ready to evaluate a new, unseen data point. For a new instance, the system first uses the optimal reliability models to predict an instance-specific reliability score for every potential judge in the pool. It then applies the *Jury Selection* and *Dynamic Score Aggregation* mechanisms to select the top $K$ judges and compute the final, weighted score.

## 4 Experimental Setup

To validate the effectiveness of our LLM Jury-on-Demand framework, we conducted a series of experiments designed to measure its performance against standard evaluation methods. This section details the datasets, evaluation protocol, and implementation specifics of our experiments.

### 4.1 Datasets and Tasks

Our evaluation spans two challenging natural language generation tasks: summarization and retrieval-augmented generation (RAG). For each task, we focus on evaluating three core metrics: groundedness, relevance, and completeness. These metrics are essential for assessing the quality and trustworthiness of generated text. Detailed definitions for each metric are provided in Appendix B.

To train our jury framework, we chose datasets with human annotations for these dimensions. We reviewed a diverse set of datasets for training, selecting 3-4 datasets per task by metric dimension to ensure coverage across various domains. To prevent any single dataset or any single metrics category from dominating a particular evaluation task, we applied stratified down-sampling where necessary. The details are provided in Appendix C.

### 4.2 Evaluation Protocol

Our experimental protocol is designed to ensure a fair and rigorous comparison between our proposed system and relevant baselines.

**Evaluation Metric.** The primary metric for our experiments is the Kendall's Tau correlation coefficient (KENDALL, 1938). This non-parametric statistic measures the ordinal association between two sets of rankings. In our context, it quantifies how well a system's

evaluation scores align with the rankings provided by human experts. A Kendall's Tau value close to 1 indicates strong agreement with human judgment.

**Judge Prompting Protocol.** To generate the raw scores for our experiment, each potential judge model was prompted with a carefully structured template. This template includes a system prompt to set the judge's persona ("You are a helpful, respectful and honest assistant") and a user prompt that defines the task, the specific metric (e.g., Completeness), and the required scoring format. The scoring scale was adapted to the task: for summarization, all judges were instructed to provide a single integer score from 1 (lowest quality) to 5 (highest quality); for RAG, a scale of 1 (lowest quality) to 3 (highest quality) was used. The full prompt structure is provided in the Appendix D.

**Baselines.** To benchmark our system's performance, we established a judge pool consisting of 9 diverse LLMs. This pool serves as the foundation for all evaluation methods compared in our study and includes the following models: Claude 3.7 SONNET (Cla), Gemini 2.5 Pro (Comanici et al., 2025), Gemini 2.5 Flash, Gemini 2.0 Flash, GPT-OSS-20B (Agarwal et al., 2025), GPT-OSS-120B, Gemma 3-27B-IT (Team et al., 2025), Phi-4-reasoning (Abdin et al., 2025), and LLAMA-3.2-3B-Instruct (Grattafiori et al., 2024). From this pool, we formed two categories of baselines:

1. **Single-Judge Baselines:** The performance of each of the 9 judges when used as a standalone evaluator.
2. **Static-Jury Baselines:** We compare against four distinct static jury formulations to rigorously test the benefits of dynamic selection:
   - **Static Jury (Average-All):** The performance of a non-adaptive, naive jury that uses all 9 judges in the pool. For each data point, the final score for this baseline is the simple average of the raw scores from all 9 judges.
   - **Static Jury (Average-Top-K):** This baseline identifies the Top-$K$ best-performing single judges based on their Kendall's Tau on the validation set. The final score is the simple average of these Top-$K$ judges. The value of $K$ is tuned on the validation set.
   - **Static Jury (Weighted-Regression):** A regression-based jury using all 9 judges. We train a linear regression model without intercept using human annotation scores as labels and single judge scores as features on the training set.
   - **Static Jury (Weighted-Tau):** A performance-weighted average of all 9 judges. The weights are derived from each judge's validation Kendall's Tau, normalized using a softmax function.

### 4.3 IMPLEMENTATION DETAILS

This section provides the specific procedures used to configure and train our system for the experiments.

First, we prepared the data for each task and metric by combining all relevant source datasets. For example, to evaluate the completeness metric for the summarization task, we aggregated the data from SummEval, TLDR, and UniSumEval into a single, combined dataset. We then performed a global 60-20-20 split on this combined data to create our final training, validation, and holdout test sets, ensuring data from all sources were represented in each split. As described in Sec. 3.3, all human and model scores were then normalized to a $[0, 1]$ range to ensure consistency.

Next, we determined the system's optimal configuration through a comprehensive hyperparameter tuning process on the combined validation set, as outlined in Sec. 3.5. We defined a search space for three key hyperparameter categories: the jury size $K$ (ranging from 2 to 8), the per-judge tolerance values $\tau$ used for training the reliability predictors, and the internal parameters of the XGBoost models. We exclude $K = 1$ as it reduces the jury to a single judge selector, effectively duplicating the Single-Judge baseline paradigm. We also exclude $K = 9$ (the full pool), as this configuration represents a static ensemble identical to the Average-All baseline, negating the benefit of dynamic selection. The sets of tolerance values

were chosen to reflect the varying scales of the original human annotation scores across the different source datasets. For each candidate configuration, we evaluated its performance by applying the full per-instance jury selection and weighting pipeline (Sec. 3.4) to every data point in the combined validation set. The final optimal configuration was chosen based on which set of hyperparameters yielded the highest Kendall's Tau correlation on this combined validation set. This finalized configuration was then used for the final, unbiased evaluation on the locked-away test set.

Additionally, to assess robustness beyond validation-based tuning, we conduct ablation studies on jury size ($K$) and tolerance ($\tau$), detailed in Appendix H.2 and H.3. Jury size is varied from 1 to 9 across two representative tasks, Summarization-Completeness and RAG-Groundedness, over 10 independent runs. As shown in Fig. 10, performance trends differ by task: for Summarization-Completeness, accuracy improves initially, peaks at $K = 3$, and then declines as weaker judges are added, diluting decision quality. In contrast, RAG-Groundedness benefits from larger juries, where adding more judges enhances robustness against noisy predictions.

## 5 Results and Analysis

This section evaluates our LLM Jury-on-Demand framework against the single-judge and static-jury baselines. To ensure robustness, all experiments were repeated 10 times with different random seeds for data partitioning, and we report the mean and standard deviation of the results. Our analysis presents the main performance, both overall and at a granular dataset-level, followed by a diagnostic analysis of our system's internal mechanics, including feature importance and judge selection frequency.

### 5.1 Performance Analysis

We assess performance by comparing the Kendall's Tau correlation of each method with human judgment on the test sets for each task-metric combination. The distribution of results from our 10 independent runs is summarized in Fig. 2.

The results clearly demonstrate that our Jury-on-Demand framework consistently outperforms all baselines across every task and metric. In all six categories, our method achieves the highest mean correlation with human judgment, validating its effectiveness. For instance, in the challenging RAG-Groundedness task, our system achieves a mean Kendall's Tau of 0.68 ($\pm 0.02$). This represents a significant improvement not only over the simple Static Jury (Average-All) (0.61 $\pm 0.02$) but also over the stronger optimized baselines, such as Static Jury (Weighted-Regression) (0.62 $\pm 0.02$) and the strongest single judge for that task, GPT-OSS-120B (0.63 $\pm 0.01$).

To provide a more detailed view of performance, we now analyze the results for the RAG-Groundedness task at the individual dataset level. The granular results in Table 1 confirm the findings from the overall analysis: our Jury-on-Demand system outperforms all four static jury baselines and the best-performing single judge on every individual dataset for this task. The performance lift is particularly pronounced on the CAQA dataset, indicating that our system is robust and its advantages are not an artifact of data aggregation but hold true at a more granular level. The full breakdown of results for all tasks and datasets is available in Appendix E.1. We also report statistical significance and effect sizes. Specifically, we perform one-sided Wilcoxon signed-rank tests (Wilcoxon, 1945) and compute Cliff's delta (Cliff, 1993), a non-parametric effect size metric that quantifies the difference between two groups. The results are presented in Appendix E.1.

Finally, the results also highlight the inherent unreliability of relying on a single LLM as an evaluator. As shown in Fig. 2, the performance of single judges is highly variable. The "best" single judge changes from one task to another; for example, Claude 3.7 SONNET is often the strongest single judge for Summarization-Groundedness, but it is one of the weakest for Summarization-Completeness. This instability proves that there is no single "best" LLM judge, making a static choice of evaluator a risky and unreliable strategy.

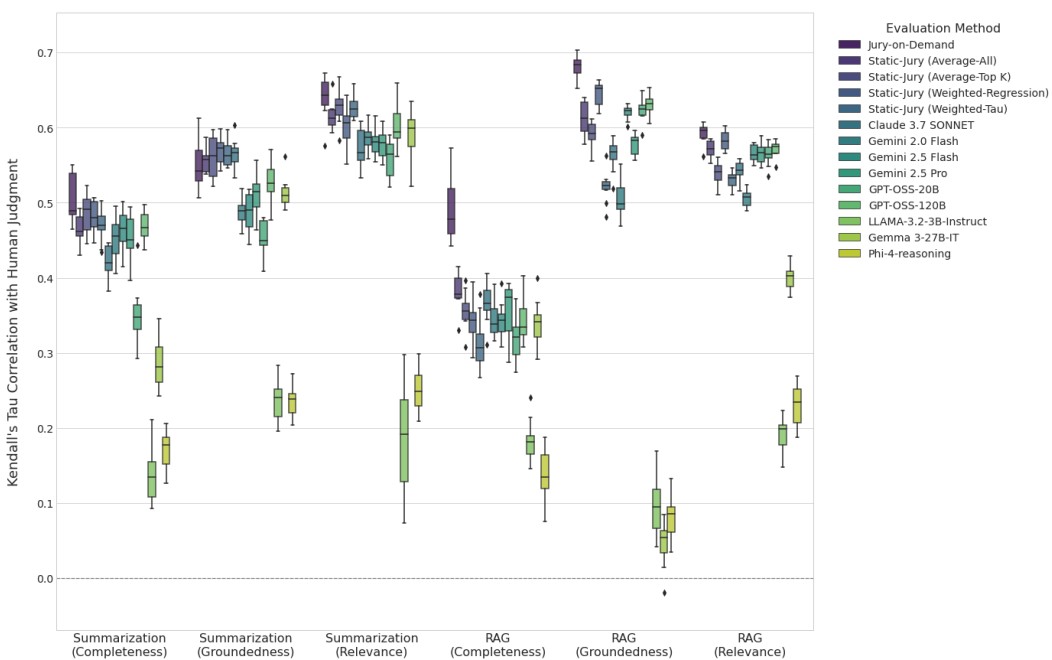

Figure 2: Overall performance comparison over 10 runs. Boxplot of Kendall's Tau correlation between each evaluation method's scores and human judgements, aggregated across all datasets for the 6 task-metric combinations. Our Jury-on-Demand system achieves the highest median correlation in nearly all categories and shows the most robust performance.

The comparison between our Jury-on-Demand and the various static jury baselines isolates the benefit of our core contribution. While baselines like Static Jury (Average-TopK) are competitive, our dynamic system is consistently superior. This demonstrates that the primary performance gain comes not just from using a jury, but from the ability to dynamically select and weight its members based on the context of the input. The stability of this outperformance, demonstrated across 10 runs, underscores the reliability of our dynamic approach.

Table 1: Granular Performance on RAG-Groundedness. Mean Kendall's Tau correlation (± std. dev.) on the individual test sets across 10 runs. Our Jury-on-Demand system consistently outperforms all static baselines and the best single judge.

| Dataset | Jury-on-Demand | Static (Avg-All) | Static (Avg-TopK) | Static (W-Reg) | Static (W-Tau) | Best Single |
|---------|----------------|------------------|-------------------|----------------|----------------|-------------|
| CAQA | $0.67 \pm 0.03$ | $0.62 \pm 0.02$ | $0.60 \pm 0.03$ | $0.62 \pm 0.02$ | $0.58 \pm 0.03$ | $0.60 \pm 0.03$ (GPT-OSS-120B) |
| HaluEval | $0.77 \pm 0.02$ | $0.73 \pm 0.02$ | $0.74 \pm 0.03$ | $0.73 \pm 0.02$ | $0.57 \pm 0.04$ | $0.77 \pm 0.02$ (GPT-OSS-120B) |
| RagTruth | $0.58 \pm 0.04$ | $0.54 \pm 0.05$ | $0.34 \pm 0.06$ | $0.55 \pm 0.02$ | $0.18 \pm 0.05$ | $0.56 \pm 0.04$ (Gemini 2.5 Flash) |

## 5.2 ANALYZING THE INTERACTION BETWEEN JUDGES, TASKS, AND DATA ATTRIBUTES

We now analyze judge selection patterns within juries across tasks and datasets. Fig. 3 summarizes the selection frequency of each judge for summarization, groundedness, and

completeness tasks. The results reveal distinct preferences: Claude 3.7 Sonnet and Gemini 2.5 Pro are frequently selected for groundedness evaluation but are rarely chosen for completeness. In contrast, Gemini 2.5 Flash, Gemini 2.0 Flash and GPT-OSS-120B are commonly selected for completeness yet seldom appear in groundedness evaluations. A comprehensive comparison of judge selection across all tasks is in Appendix E.2 along with the analysis of judge selection frequencies for individual datasets within the task.

To further explore how data properties influence judge selection, we examine selection patterns across different bins of key attributes, using CAQA from the RAG-Groundedness task as a case study. The jury for this task comprises eight judges, and for clarity, Fig. 4 highlights the top three selected judges. The results reveal distinct trends: Compared to Gemini models, two GPT models are more frequently selected when the source context compression ratio is low (below 0.75), whereas Claude 3.7 Sonnet is more often selected when the compression ratio is high. Additionally, Llama and Phi models also show increased selection frequency under high compression ratios.

These findings reinforce the importance of constructing dynamic juries that adapt to specific data characteristics and demonstrate the potential of predicting judge reliability based on interpretable data properties.

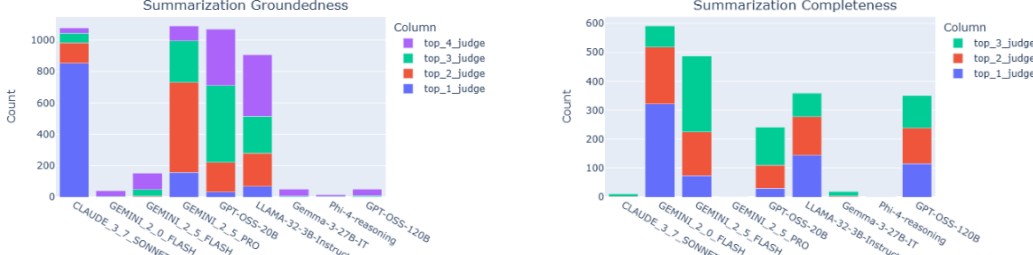

Figure 3: Selection frequency of the judge in the jury. Top k judge means that the judge has the k-th highest reliability score in the jury. Claude 3.7 Sonnet and Gemini 2.5 Pro are favored in groundedness, while Gemini 2.0 Flash, Gemini 2.5 Flash and GPT-OSS-120B are more often selected for completeness.

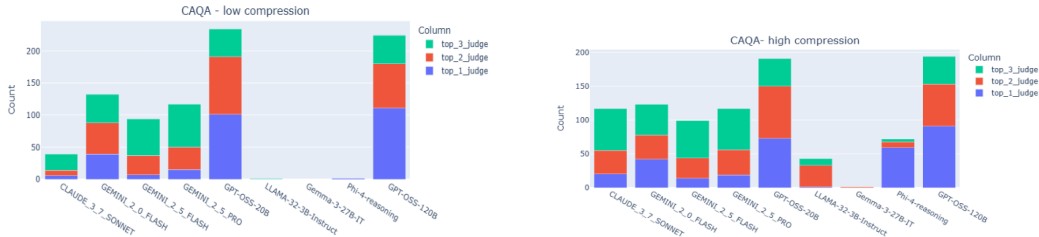

Figure 4: Judge selection variation in RAG-Groundedness (CAQA) by compression ratio: GPT models are favored when compression is low; Claude 3.7 Sonnet, Llama, and Phi are more frequently selected when compression is high.

Table 2 presents the top five most important features for the judge reliability XGBoost model, as determined by permutation feature importance (Fisher et al., 2019), with the Summarization-Groundedness task as an example. For illustration, we focus on Gemini 2.5 Pro and GPT-OSS-120B, while the complete results for all judges are in Appendix E.2. The analysis reveals substantial variation in feature importance across judges. For instance, embedding-related features are more influential for Gemini 2.5 Pro, suggesting that different judges rely on distinct data properties when assessing reliability. We further aggregate the top five features that frequently appear across tasks, with results provided in Appendix E.2. The analysis reveals clear task-specific trends: text size-related features, such as word count and compression ratio, along with token entropy, are more prominent in RAG tasks. In

contrast, embedding-based features, including PCA components and embedding similarity, play a more significant role in summarization tasks. These findings align with the ablation analysis in Appendix H.1, which shows that removing embedding features leads to a greater performance drop in the summarization task compared to RAG. These observations imply that evaluation reliability is task-dependent and further demonstrate that our approach effectively links data characteristics to judge reliability, enabling more informed and adaptive jury construction across diverse evaluation scenarios.

Table 2: Top 5 important features for the Summarization-Groundedness task. Gemini 2.5 Pro relies more on embedding-based features.

| Judge | Feature 1 | Feature 2 | Feature 3 | Feature 4 | Feature 5 |
|-------|-----------|-----------|-----------|-----------|-----------|
| Gemini 2.5 Pro | output pca2 | input pca1 | input negation sentence | input pca9 | output embedding education |
| GPT-OSS-120B | output reading index | input embedding business | output ngram repetition | output embedding business | input modality |

## 6 CONCLUSION

In this work, we addressed the critical challenge of creating scalable and reliable evaluation systems for Large Language Models. We introduced LLM Jury-on-Demand, a novel framework that moves beyond the static aggregation methods of prior jury-based systems by learning to predict judge reliability and dynamically assembling an expert jury for each data point. Our experimental results demonstrated that this adaptive approach consistently outperforms both single-judge and static-jury baselines in aligning with human expert judgement. This confirms our central hypothesis that for automated evaluation to be trustworthy, it must be context-aware and adaptive, rather than static.

While our results are promising, this work has several limitations that open clear paths for future research. Our current framework relies on a human-annotated dataset to train the reliability predictors; future work could explore semi-supervised or self-supervised techniques to reduce this dependency and enhance scalability. Furthermore, we conduct experiments to assess the framework's ability to generalize beyond its training domains by training on a subset of domains and applying it to held-out domains. The results indicate partial generalization: certain learned patterns transfer effectively to new domains, while others do not (see Appendix I for details). These findings suggest that the framework's generalizability is contingent on both the diversity of the training data and the characteristics of the unseen domains. As additional annotated data becomes available and incorporated into training, we anticipate that the framework's capacity to generalize will improve, enabling broader applicability across diverse areas.

Another promising direction for future work is mitigating bias in judge scores. For example, certain judges may consistently assign higher or lower scores compared to human annotations. We explore score calibration in Appendix J. Our experiments show that calibration can sometimes improve alignment between judge scores and human annotations, but in other cases, it may amplify the bias. Addressing this challenge remains an open problem, and we plan to investigate more robust approaches in future work.

## ETHICS STATEMENT

This work leverages large language models (LLMs) to evaluate content generated by LLMs. All data used in this study are publicly available, and the evaluation focuses on groundedness, completeness, and relevance. Based on these factors, we do not anticipate any ethical concerns associated with this work.

## Reproducibility Statement

We provide a detailed description of our methodological framework in Section 3. Implementation details are outlined in Section 4.3, including the hyperparameter search range for XGBoost, which is specified in Section K. Information about the datasets, text data features, and the prompts used can be found in Appendix C, A, and D, respectively.

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

## A  LIST OF DATA FEATURES

Table 3: Data features in judge reliability model.

| Feature Name | Explanation | Category |
|---|---|---|
| COUNT_WORD | Number of words in the context. | Text size |
| COUNT_CHAR | Number of characters in the context. | Text size |
| COUNT_SENTENCE | Number of sentences in the context. | Text size |
| COUNT_PARAGRAPH | Number of paragraphs in the context. | Text size |
| CHAR_COMPRESSION | The ratio of the number of characters in the output context (summary or answer) to those in the input context (article or cited context). | Text size |
| WORD_COMPRESSION | The ratio of the number of words in the output context (summary or answer) to those in the input context (article or cited context). | Text size |
| NUM_WORD_SENTENCE | Average number of words per sentence. | Text size |
| NUM_CHAR_WORD | Average number of characters per word. | Text size |
| DIFFICULT_WORD | Number of difficult words in the context, defined as words with more than two syllables. | Special words |
| STOP_WORDS | Number of stop words in the context, such as "I", "to", "and", "of", etc. | Special words |
| MODALITY | Number of modality verbs in the context, such as "can", "could", "should", etc. | Special words |
| NUMBER_COUNT | Count of numbers in the context, such as date, time, percent etc. | Special words |
| NAMED_ENTITY | Count of named entities in the context, including person, organization, date, time, Geo-Political entity, location and money. | Special words |
| FACTUAL_DENSISTY | Number of entities divided by context length. | Special words |
| NGRAM_COUNT | Count of n(3)-grams in the context. | Special words |
| NEGATION_SENTENCE | Count of sentences with negation words such as "no", "not", "never", etc. | Special words |
| COUNT_QUESTION | Number of questions in the context. | Special words |
| TOKEN_ENTROPY | Shannon entropy on token distribution. | Text complexity |
| LEXICAL_DIVERSITY | Number of unique words divided by the total number of words in the context. | Text complexity |
| READING_INDEX | Flesch reading ease index, measuring the difficulty of reading the context. | Text complexity |
| NGRAM_REPETITION | N(3)-gram repetition ratio in the context. | Text complexity |
| SENTENCE_SIMILARITY | Average cosine similarity between each pair of sentence embeddings. | Text complexity |

| Feature Name | Explanation | Category |
|---|---|---|
| SYNTACTIC_ AMBI-GUITY | The average number of syntactically ambiguous POS tags (IN, TO) across sentences. These tags indicate structural complexity or multiple possible parses. | Text complexity |
| SEMANTIC_ AMBIGU-ITY | The average number of WordNet senses per word across all sentences in the text. A higher average suggests more potential meanings and interpretive complexity. | Text complexity |
| COREFERENCE_CHAIN | The average number of pronouns per sentence. | Text complexity |
| COREFERENCE_ AM-BIGUOUS | Number of pronoun-ambiguous sentences in the context. Sentences with more than one pronoun are classified as ambiguous. | Text complexity |
| SYNTACTIC_ANOMALY | Number of syntactic anomaly sentences in the context. A sentence is syntactic anomaly if either subject or verb is missing, or both are missing. | Text complexity |
| RHETORICAL_ STRUCTURE | Number of sentences with discourse markers (however, therefore) and rhetorical structure (moreover, in contrast, thus, instead, etc.). | Text complexity |
| POLARITY | The polarity score of the context, measuring the emotional tone of the text. | Text complexity |
| SUBJECTIVITY | The subjectivity score of the context, measuring the degree of personal opinion or factuality. | Text complexity |
| PCA | Text embeddings are computed via mean pooling and reduced in dimensionality using PCA. The top 10 principal components are used as features. | Embedding |
| Topic similarity | Cosine similarity between each text's embedding and a set of predefined topic embeddings - market, bank, business, tech, education, politics, legal, sports, media, science. | Embedding |

# B  Evaluation Metric Definitions

For each task, we focus on evaluating three core metrics: groundedness, relevance, and completeness.

**Groundedness:** Assesses how well the output is supported by the context of the input. A grounded output accurately reflects the source information without introducing unsupported or fabricated content. This dimension is closely related to the concept of hallucination in language models.

**Relevance:** Measures the degree to which the output includes only essential and contextually appropriate information, avoiding extraneous or off-topic content. However, for RAG, annotated data which assesses output (answer) relevance is not readily available, so instead we check retrieval relevance. Specifically, how closely and thoroughly the retrieved context addresses the posed question. A context is considered relevant if it is clearly focused on the

question and provides sufficient information to support a complete and accurate answer. Similarly, we can assess how relevant the context is with respect to the reference answer.

**Completeness:** Captures whether the output includes all critical information from the input context, ensuring comprehensive coverage.

## C  LIST OF DATASETS

The datasets used for different evaluation metrics are listed in Table 4 (summarization) and Table 5 (RAG). We prioritized datasets with annotated scores for completeness, grounded-ness, or relevance. However, annotated data for the completeness metric is relatively scarce. To address this, we simulate incomplete outputs by removing sentences from multi-sentence references and assigning scores accordingly. This approach is applied to the SummEval dataset for summarization and to all three datasets used in the RAG task.

Table 4: List of Datasets for Summarization

| Metric | Data | Size | Annotation (Ann.) | Ann. scale | Domain |
|---|---|---|---|---|---|
| Completeness | TL;DR (Stiennon et al., 2020) | 1680 | Coverage score measures how much important information from the original post is covered | 1 to 7 | Reddit Discussion |
| | UniSumEval (Lee et al., 2024) | 1623 | Completeness ratio measures the proportion of key facts inferable from the summary | 0 to 1 (fraction) | Nine different domains including Wikihow, CNN/DM, GovReport, PubMed, etc. |
| | SummEval (Fabbri et al., 2021) | 793 | Original data does not have completeness score. To create annotation for completeness, we assign score 5 to the reference summary and remove different proportions of sentences in the reference summary to create incomplete summaries with scores 1 to 4 | 1 to 5 | CNN/DM |
| Groundedness | SummEval (Fabbri et al., 2021) | 1600 | Consistency score measures the factual alignment between the summary and the summarized source. A factually consistent summary contains only statements that are entailed by the source document. | 1 to 5 | CNN/DM |
| | TL;DR (Stiennon et al., 2020) | 1597 | Accuracy score, measures to what degree the statements in the summary are stated in the post | 1 to 7 | Reddit Discussion |

Table 4: List of Datasets for Summarization

| Metric | Data | Size | Annotation (Ann.) | Ann. scale | Domain |
|---|---|---|---|---|---|
| | DialSummEval (Gao & Wan, 2022) | 1400 | Consistency score measures how well the summary aligns with the dialogue in fact. It focuses on whether the summary contains factual errors. | 1 to 5 | SAMSum (message dialogues) |
| | UniSumEval (Lee et al., 2024) | 1624 | Faithfulness score measures the proportion of factually correct summary sentences. | 0 to 1 (fraction) | 9 different domains including Wikihow, CNN/DM, GovReport,PubMed, etc. |
| Relevance | OpinSummEval (Shen & Wan, 2023) | 1400 | Aspect relevance, measures whether the mainly discussed aspects in the reviews are covered exactly by the summary. It focuses on whether summary correctly reflects the main aspects in the reviews. | 1 to 5 | Yelp reviews |
| | SummEval (Fabbri et al., 2021) | 1600 | Relevance score, measures selection of important content from the source. The summary should include only important information from the source document. | 1 to 5 | CNN/DM |
| | DialSummEval (Gao & Wan, 2022) | 1400 | Relevance score, measures how well the summary captures the key points of the dialogue. It focuses on whether all and only the important aspects are contained in the summary. | 1 to 5 | SAMSum (message dialogues) |
| | UniSumEval (Lee et al., 2024) | 1623 | Conciseness score, measures the proportion of summary sentences aligned with the key-facts. | 0 to 1 (fraction) | Nine different domains including Wikihow, CNN/DM, GovReport,PubMed, etc. |

Table 5: List of Datasets for RAG

| Metric | Data | Size | Annotation (Ann.) | Ann. scale | Domain |
|---|---|---|---|---|---|
| Completeness | ASQA (Stelmakh et al., 2022) | 3231 | Original data does not have completeness score. To create annotations for completeness, we assign score 2 to the reference answer and remove different proportions of sentences in the answer to create incomplete answers with scores 0 and 1 | 0, 1, 2 | Wikipedia |
| | ALCE (Gao et al., 2023) | 593 | Original data does not have completeness score. To create annotations for completeness, we assign score 2 to the reference answer and remove different proportions of sentences in the answer to create incomplete answers with scores 0 and 2 | 0, 1, 2 | Wikipedia and Reddit |
| | QASPER (Dasigi et al., 2021) | 561 | Original data does not have completeness score. To create annotations for completeness, we assign score 2 to the reference answer and remove different proportions of sentences in the answer to create incomplete answers with scores 0 and 3 | 0, 1, 2 | NLP research papers |
| Groundedness | RagTruth (Niu et al., 2024) | 3206 | We assigned scores 0 to 2 based on count of hallucination spans in the output | 0, 1, 2 | MS Marco |
| | HaluEval (Li et al., 2023) | 3000 | Whether output contained hallucinated content | 0, 1 | HotPot-QA |
| | CAQA (Hu et al., 2025) | 3000 | Whether the cited text supports the answer. There are 4 labels, supportive, partially supportive, contradict and irrelevant | 0,1,2 | Knowledge Graph generated questions |
| Relevance | MS MARCO (Bajaj et al., 2016) | 3200 | Whether the cited text is relevant to the question | 0, 1, 2, 3 | Bing queries |

Table 5: List of Datasets for RAG

| Metric | Data | Size | Annotation (Ann.) | Ann. scale | Domain |
|--------|------|------|-------------------|-----------|--------|
| | HotpotQA (Yang et al., 2018) | 3000 | Whether the cited text is relevant to the question | 0,1,2 | Wikipedia-based QA |
| | ALCE (Gao et al., 2023) | 2896 | Whether the cited text is relevant to the answer | 0,1,2 | Wikipedia and Reddit |

## D  PROMPT TEMPLATE

We use the same prompt template across tasks. Below we list the prompt for summarization (Table 6) and RAG (Table 7), with dimension groundedness.

Table 6: Prompt for summarization.

System prompt:
You are a helpful, respectful and honest assistant. Follow instructions attentively. Do not add any extraneous information.

User prompt:
You will be given an input context followed by a corresponding summary. Your task is to evaluate the quality of the summary with respect to groundedness.

Definition of Groundedness: Groundedness refers to how well the summary is supported by the content of the input context. A grounded summary should accurately reflect the information presented in the input context without introducing unsupported claims.

Provide a score for this summary on a scale from 1 (worst) to 5 (best). Valid scores are: 1, 2, 3, 4, or 5.

Output format:
[score number](on its own line, only one number here, no brackets or letters or 'score:')
[explanation](starting on the next line)

Conversation: source
Summary: output

Table 7: Prompt for RAG.

| |
| --- |
| System prompt:
You are a helpful, respectful and honest assistant. Follow instructions attentively. Do not add any extraneous information. |
| User prompt:
You will be given a question ('Question' below) followed by a response ('Response' below) for the question. After that, cited background information is provided ('Context' below). The response was generated by a LLM based on the cited background information. Your task is to evaluate the quality of the response with respect to groundedness.

Definition of Groundedness: Groundedness refers to how well the response is supported by the content of the cited background information. A grounded response should accurately reflect the cited background information without introducing unsupported claims.

Provide a score for the response on a scale of 0 (bad), 1 (fair), or 2 (good). Valid scores are: 0, 1, or 2.

Output format:
[score number](on its own line, only one number here, no brackets or letters or 'score:')
[explanation](starting on the next line)

Question: question
Response: response
Context: cited text |

## E  FULL EXPERIMENTAL RESULTS

We first show the jury performance, then analyze the interactions between judges, tasks and data properties.

### E.1  FULL RESULTS BY TASK AND DATASET

Tables 8 - 11 present the complete Kendall's Tau correlation results of our experiments. Due to the table size limit, we split the results for clarity. Tables 8 and 10 compare our Jury-on-Demand system against the four Static Jury baselines (Average-All, Average-TopK, Weighted-Regression, and Weighted-Tau). Tables 9 and 11 compare Jury-on-Demand against the 9 Single-Judge baselines. Each row corresponds to a specific evaluation set, either an "Overall" aggregation or an individual source dataset. All values represent the mean Kendall's Tau rank correlation coefficient ($\pm$ standard deviation) calculated across 10 independent runs. Higher values indicate better performance and stronger alignment with human judgment.

We also report statistical significance and effect sizes. Specifically, we perform one-sided Wilcoxon signed-rank tests (Wilcoxon, 1945) to compare the Tau differences between Jury-on-Demand and either static juries or single judges. The corresponding p-values are presented in Table 12 through Table 15. Among these p-values, 82% are statistically significant ($p < 0.05$). Values in parentheses represent Cliff's delta (Cliff, 1993), a non-parametric effect size metric that quantifies the difference between two groups, with Jury-on-Demand serving as the baseline. According to conventional thresholds, an effect size is considered large if it exceeds 0.47 and medium if it falls between 0.33 and 0.47. Across all Cliff's delta values, 82% are classified as either large (72%) or medium (10%). This high proportion of significant p-values and substantial effect sizes indicates that, in most cases, Jury-on-Demand outperforms static baselines and single judges.

Table 8: Summarization Results: Jury-on-Demand vs Static Jury baselines. Numbers in parentheses are standard deviation.

| Data | Jury-on-Demand | Static Jury (Average-All) | Static Jury (Average-TopK) | Static Jury (Weighted-Regression) | Static Jury (Weighted-Tau) |
|------|------|------|------|------|------|
| **Completeness** | | | | | |
| Overall | 0.51 | 0.46 | 0.49 | 0.48 | 0.47 |
| | (0.03) | (0.02) | (0.03) | (0.02) | (0.02) |
| SummEval | 0.74 | 0.62 | 0.68 | 0.68 | 0.61 |
| | (0.05) | (0.05) | (0.07) | (0.06) | (0.04) |
| TL;DR | 0.39 | 0.39 | 0.45 | 0.43 | 0.40 |
| | (0.06) | (0.03) | (0.04) | (0.03) | (0.03) |
| UniSumEval | 0.68 | 0.61 | 0.63 | 0.58 | 0.62 |
| | (0.05) | (0.05) | (0.05) | (0.05) | (0.05) |
| **Groundedness** | | | | | |
| Overall | 0.55 | 0.56 | 0.56 | 0.57 | 0.57 |
| | (0.04) | (0.02) | (0.03) | (0.02) | (0.02) |
| SummEval | 0.62 | 0.65 | 0.62 | 0.66 | 0.65 |
| | (0.05) | (0.04) | (0.05) | (0.03) | (0.03) |
| TL;DR | 0.42 | 0.46 | 0.46 | 0.46 | 0.46 |
| | (0.07) | (0.05) | (0.04) | (0.04) | (0.05) |
| UniSumEval | 0.61 | 0.62 | 0.63 | 0.64 | 0.63 |
| | (0.07) | (0.07) | (0.07) | (0.06) | (0.06) |
| DialSummEval | 0.71 | 0.66 | 0.69 | 0.69 | 0.68 |
| | (0.03) | (0.02) | (0.03) | (0.03) | (0.02) |
| **Relevance** | | | | | |
| Overall | 0.64 | 0.62 | 0.63 | 0.60 | 0.63 |
| | (0.03) | (0.02) | (0.02) | (0.03) | (0.02) |
| SummEval | 0.71 | 0.69 | 0.71 | 0.69 | 0.71 |
| | (0.08) | (0.03) | (0.06) | (0.07) | (0.03) |
| DialSummEval | 0.69 | 0.65 | 0.63 | 0.64 | 0.65 |
| | (0.03) | (0.02) | (0.04) | (0.03) | (0.03) |
| OpinSummEval | 0.44 | 0.43 | 0.42 | 0.43 | 0.44 |
| | (0.06) | (0.04) | (0.05) | (0.05) | (0.05) |
| UniSumEval | 0.45 | 0.40 | 0.39 | 0.37 | 0.42 |
| | (0.09) | (0.10) | (0.11) | (0.09) | (0.10) |

Table 9: Summarization Results: Jury-on-Demand vs Single Judge baselines. Numbers in parentheses are standard deviation. Here Gemn. is Gemini and LL is LLAMA.

| Data | Jury-on-De-mand | Claude 3.7 | Gemn. 2.0 | Gemn. 2.5 | Gemn. 2.5 Pro | GPT - OSS - 20B | GPT - OSS - 120B | LL - 3.2 | Gemma 3 | Phi - 4 |
|------|------|------|------|------|------|------|------|------|------|------|
| **Completeness** | | | | | | | | | | |
| Overall | 0.51 | 0.42 | 0.45 | 0.46 | 0.45 | 0.35 | 0.47 | 0.14 | 0.29 | 0.17 |
| | (0.03) | (0.02) | (0.03) | (0.03) | (0.03) | (0.04) | (0.02) | (0.04) | (0.03) | (0.03) |
| Summ - Eval | 0.74 | 0.54 | 0.63 | 0.68 | 0.57 | -0.09 | 0.66 | 0.12 | 0.31 | 0.26 |
| | (0.05) | (0.11) | (0.07) | (0.06) | (0.10) | (0.20) | (0.07) | (0.11) | (0.16) | (0.09) |
| TL;DR | 0.39 | 0.38 | 0.38 | 0.42 | 0.44 | 0.34 | 0.40 | 0.11 | 0.13 | -0.02 |
| | (0.06) | (0.05) | (0.06) | (0.05) | (0.05) | (0.05) | (0.04) | (0.05) | (0.05) | (0.06) |
| UniSum - Eval | 0.68 | 0.50 | 0.54 | 0.59 | 0.53 | 0.63 | 0.66 | 0.28 | 0.56 | 0.28 |

| Data | Jury-on-Demand | Claude 3.7 | Gemn. 2.0 | Gemn. 2.5 | Gemn. 2.5 Pro | GPT-OSS-20B | GPT-OSS-120B | LL-3.2 | Gemma 3 | Phi-4 |
|---|---|---|---|---|---|---|---|---|---|---|
| | (0.05) | (0.05) | (0.06) | (0.05) | (0.06) | (0.03) | (0.04) | (0.09) | (0.04) | (0.06) |
| **Groundedness** | | | | | | | | | | |
| Overall | 0.55 | 0.57 | 0.49 | 0.49 | 0.51 | 0.45 | 0.53 | 0.23 | 0.51 | 0.24 |
| | (0.04) | (0.02) | (0.02) | (0.03) | (0.03) | (0.02) | (0.03) | (0.03) | (0.02) | (0.02) |
| Summ-Eval | 0.62 | 0.63 | 0.55 | 0.62 | 0.64 | 0.56 | 0.60 | 0.20 | 0.59 | 0.17 |
| | (0.05) | (0.05) | (0.06) | (0.04) | (0.04) | (0.07) | (0.04) | (0.09) | (0.05) | (0.09) |
| TL;DR | 0.42 | 0.40 | 0.42 | 0.40 | 0.42 | 0.29 | 0.41 | 0.06 | 0.42 | 0.12 |
| | (0.07) | (0.03) | (0.05) | (0.05) | (0.05) | (0.05) | (0.05) | (0.04) | (0.04) | (0.06) |
| UniSum-Eval | 0.61 | 0.62 | 0.58 | 0.61 | 0.64 | 0.62 | 0.57 | 0.12 | 0.56 | 0.31 |
| | (0.07) | (0.07) | (0.07) | (0.08) | (0.06) | (0.09) | (0.09) | (0.17) | (0.08) | (0.10) |
| Dial-Summ-Eval | 0.71 | 0.59 | 0.63 | 0.71 | 0.70 | 0.65 | 0.67 | 0.33 | 0.63 | 0.25 |
| | (0.03) | (0.04) | (0.06) | (0.03) | (0.04) | (0.05) | (0.03) | (0.06) | (0.04) | (0.06) |
| **Relevance** | | | | | | | | | | |
| Overall | 0.64 | 0.57 | 0.59 | 0.58 | 0.58 | 0.56 | 0.60 | 0.19 | 0.59 | 0.25 |
| | (0.03) | (0.03) | (0.02) | (0.02) | (0.02) | (0.02) | (0.03) | (0.07) | (0.04) | (0.03) |
| Dial-Summ-Eval | 0.69 | 0.51 | 0.57 | 0.58 | 0.51 | 0.66 | 0.68 | 0.22 | 0.59 | 0.43 |
| | (0.03) | (0.08) | (0.06) | (0.04) | (0.04) | (0.04) | (0.03) | (0.09) | (0.04) | (0.05) |
| Opin-Summ-Eval | 0.44 | 0.34 | 0.39 | 0.32 | 0.35 | 0.37 | 0.37 | 0.20 | 0.40 | 0.18 |
| | (0.06) | (0.07) | (0.04) | (0.06) | (0.05) | (0.07) | (0.06) | (0.10) | (0.07) | (0.08) |
| Summ-Eval | 0.71 | 0.67 | 0.68 | 0.64 | 0.66 | 0.65 | 0.67 | 0.41 | 0.70 | 0.26 |
| | (0.08) | (0.06) | (0.06) | (0.09) | (0.06) | (0.05) | (0.09) | (0.16) | (0.04) | (0.08) |
| UniSum-Eval | 0.45 | 0.38 | 0.36 | 0.37 | 0.42 | 0.31 | 0.38 | 0.19 | 0.40 | 0.13 |
| | (0.09) | (0.12) | (0.10) | (0.13) | (0.10) | (0.09) | (0.11) | (0.10) | (0.10) | (0.07) |

Table 10: RAG Results: Jury-on-Demand vs Static Jury baselines. Numbers in parentheses are standard deviation.

| Data | Jury-on-Demand | Static Jury (Average-All) | Static Jury (Average-TopK) | Static Jury (Weighted-Regression) | Static Jury (Weighted-Tau) |
|---|---|---|---|---|---|
| **Completeness** | | | | | |
| Overall | 0.49 | 0.38 | 0.36 | 0.35 | 0.31 |
| | (0.04) | (0.02) | (0.02) | (0.03) | (0.03) |
| ALCE | 0.44 | 0.38 | 0.26 | 0.32 | 0.23 |
| | (0.06) | (0.08) | (0.11) | (0.09) | (0.10) |
| ASQA | 0.49 | 0.40 | 0.36 | 0.35 | 0.33 |
| | (0.06) | (0.03) | (0.03) | (0.03) | (0.06) |
| QASPER | 0.42 | 0.38 | 0.28 | 0.34 | 0.23 |
| | (0.07) | (0.12) | (0.11) | (0.08) | (0.10) |
| **Groundedness** | | | | | |
| Overall | 0.68 | 0.61 | 0.59 | 0.64 | 0.52 |
| | (0.02) | (0.02) | (0.02) | (0.02) | (0.02) |
| CAQA | 0.67 | 0.62 | 0.60 | 0.62 | 0.58 |
| | (0.03) | (0.02) | (0.03) | (0.02) | (0.03) |

| Data | Jury-on-Demand | Static Jury (Average-All) | Static Jury (Average-TopK) | Static Jury (Weighted-Regression) | Static Jury (Weighted-Tau) |
|---|---|---|---|---|---|
| HaluEval | 0.77 | 0.73 | 0.74 | 0.73 | 0.57 |
| | (0.02) | (0.02) | (0.03) | (0.02) | (0.04) |
| RagTruth | 0.58 | 0.54 | 0.34 | 0.55 | 0.18 |
| | (0.04) | (0.05) | (0.06) | (0.02) | (0.05) |
| **Relevance** | | | | | |
| Overall | 0.59 | 0.57 | 0.54 | 0.58 | 0.53 |
| | (0.01) | (0.01) | (0.02) | (0.01) | (0.01) |
| ALCE | 0.60 | 0.59 | 0.44 | 0.59 | 0.39 |
| | (0.03) | (0.03) | (0.05) | (0.04) | (0.03) |
| HotpotQA | 0.89 | 0.87 | 0.86 | 0.85 | 0.84 |
| | (0.01) | (0.01) | (0.02) | (0.02) | (0.02) |
| MS MARCO | 0.45 | 0.41 | 0.46 | 0.45 | 0.45 |
| | (0.04) | (0.05) | (0.06) | (0.05) | (0.05) |

Table 11: RAG Results: Jury-on-Demand vs Single Judge baselines. Numbers in parentheses are standard deviation. Here Gemn. is Gemini and LL is LLAMA.

| Data | Jury-on-Demand | Claude 3.7 | Gemn. 2.0 | Gemn. 2.5 | Gemn. 2.5 Pro | GPT-OSS-20B | GPT-OSS-120B | LL-3.2 | Gemma 3 | Phi-4 |
|---|---|---|---|---|---|---|---|---|---|---|
| **Completeness** | | | | | | | | | | |
| Overall | 0.49 | 0.37 | 0.34 | 0.34 | 0.36 | 0.32 | 0.34 | 0.18 | 0.34 | 0.14 |
| | (0.04) | (0.03) | (0.02) | (0.03) | (0.04) | (0.03) | (0.03) | (0.03) | (0.03) | (0.04) |
| ALCE | 0.44 | 0.38 | 0.37 | 0.29 | 0.3 | 0.42 | 0.37 | 0.13 | 0.34 | 0.07 |
| | (0.06) | (0.06) | (0.05) | (0.1) | (0.1) | (0.05) | (0.09) | (0.07) | (0.06) | (0.14) |
| ASQA | 0.49 | 0.4 | 0.37 | 0.33 | 0.32 | 0.33 | 0.38 | 0.22 | 0.33 | 0.13 |
| | (0.06) | (0.04) | (0.05) | (0.03) | (0.04) | (0.02) | (0.03) | (0.05) | (0.04) | (0.06) |
| QASPER | 0.42 | 0.3 | 0.23 | 0.39 | 0.36 | 0.37 | 0.43 | 0.07 | 0.31 | -0.03 |
| | (0.07) | (0.1) | (0.12) | (0.1) | (0.12) | (0.13) | (0.12) | (0.09) | (0.13) | (0.1) |
| **Groundedness** | | | | | | | | | | |
| Overall | 0.68 | 0.56 | 0.5 | 0.62 | 0.58 | 0.62 | 0.63 | 0.1 | 0.04 | 0.08 |
| | (0.02) | (0.02) | (0.02) | (0.01) | (0.01) | (0.02) | (0.01) | (0.04) | (0.03) | (0.03) |
| CAQA | 0.67 | 0.56 | 0.58 | 0.59 | 0.56 | 0.59 | 0.6 | 0.09 | 0 | -0.01 |
| | (0.03) | (0.02) | (0.03) | (0.03) | (0.03) | (0.03) | (0.03) | (0.05) | (0.05) | (0.06) |
| Halu-Eval | 0.77 | 0.65 | 0.51 | 0.73 | 0.72 | 0.76 | 0.77 | 0.2 | 0.22 | 0.16 |
| | (0.02) | (0.03) | (0.04) | (0.02) | (0.03) | (0.02) | (0.02) | (0.05) | (0.04) | (0.05) |
| Rag-Truth | 0.58 | 0.41 | 0.32 | 0.56 | 0.51 | 0.51 | 0.53 | 0.11 | 0.13 | 0.07 |
| | (0.04) | (0.05) | (0.04) | (0.04) | (0.04) | (0.05) | (0.05) | (0.06) | (0.06) | (0.07) |
| **Relevance** | | | | | | | | | | |
| Overall | 0.59 | 0.54 | 0.51 | 0.57 | 0.57 | 0.56 | 0.57 | 0.19 | 0.4 | 0.23 |
| | (0.01) | (0.01) | (0.01) | (0.01) | (0.01) | (0.01) | (0.01) | (0.02) | (0.02) | (0.03) |
| ALCE | 0.6 | 0.51 | 0.53 | 0.56 | 0.57 | 0.59 | 0.58 | 0.11 | 0.43 | 0.11 |
| | (0.03) | (0.04) | (0.04) | (0.04) | (0.05) | (0.03) | (0.03) | (0.05) | (0.03) | (0.04) |
| Hotpot-QA | 0.89 | 0.86 | 0.84 | 0.88 | 0.88 | 0.87 | 0.89 | 0.41 | 0.76 | 0.45 |
| | (0.01) | (0.02) | (0.02) | (0.02) | (0.02) | (0.02) | (0.01) | (0.04) | (0.02) | (0.03) |
| MS MARCO | 0.45 | 0.41 | 0.4 | 0.42 | 0.44 | 0.38 | 0.42 | 0.14 | 0.17 | 0.11 |
| | (0.04) | (0.05) | (0.03) | (0.05) | (0.05) | (0.05) | (0.05) | (0.05) | (0.05) | (0.07) |

Table 12: Summarization: p-value of Wilcoxon test for the Tau difference between Jury-On-Demand and Static Jury baselines. Numbers in paranthesis () are effect size Cliff's delta.

| Data | Static Jury (Average-All) | Static Jury (Average-TopK) | Static Jury (Weighted-Regression) | Static Jury (Weighted-Tau) |
|------|------|------|------|------|
| **Completeness** | | | | |
| Overall | 0.001 | 0.042 | 0.007 | 0.001 |
| | (0.70) | (0.26) | (0.40) | (0.68) |
| Summ - Eval | 0.001 | 0.014 | 0.019 | 0.001 |
| | (0.96) | (0.52) | (0.60) | (0.98) |
| TL;DR | 0.385 | 0.995 | 0.903 | 0.577 |
| | (0.10) | (-0.52) | (-0.26) | (-0.08) |
| UniSum - Eval | 0.001 | 0.001 | 0.001 | 0.002 |
| | (0.68) | (0.50) | (0.84) | (0.56) |
| **Groundedness** | | | | |
| Overall | 0.652 | 0.688 | 0.958 | 0.934 |
| | (-0.20) | (-0.20) | (-0.42) | (-0.36) |
| Summ - Eval | 0.947 | 0.461 | 0.935 | 0.920 |
| | (-0.22) | (0.02) | (-0.34) | (-0.24) |
| TL;DR | 0.947 | 0.993 | 0.981 | 0.981 |
| | (-0.34) | (-0.40) | (-0.36) | (-0.34) |
| UniSum - Eval | 0.947 | 0.968 | 0.995 | 0.981 |
| | (-0.24) | (-0.26) | (-0.38) | (-0.34) |
| Dial - Summ - Eval | 0.001 | 0.019 | 0.024 | 0.002 |
| | (0.84) | (0.38) | (0.42) | (0.70) |
| **Relevance** | | | | |
| Overall | 0.007 | 0.032 | 0.001 | 0.032 |
| | (0.62) | (0.40) | (0.72) | (0.50) |
| Dial - Summ - Eval | 0.001 | 0.001 | 0.001 | 0.001 |
| | (0.84) | (0.88) | (0.70) | (0.72) |
| Opin - Summ - Eval | 0.116 | 0.138 | 0.053 | 0.278 |
| | (0.20) | (0.18) | (0.28) | (0.16) |
| Summ - Eval | 0.053 | 0.461 | 0.032 | 0.161 |
| | (0.40) | (0.08) | (0.38) | (0.28) |
| UniSum - Eval | 0.042 | 0.032 | 0.019 | 0.161 |
| | (0.18) | (0.36) | (0.52) | (0.20) |

Table 13: Summarization: p-value of Wilcoxon test for the Tau difference between Jury-On-Demand and single judge. Here Gemn. is Gemini and LL is LLAMA. Numbers in paranthesis () are effect size Cliff's delta.

| Data | Claude 3.7 | Gemn. 2.0 | Gemn. 2.5 | Gemn. 2.5 Pro | GPT - OSS - 20B | GPT - OSS - 120B | LL - 3.2 | Gemma 3 | Phi - 4 |
|------|------|------|------|------|------|------|------|------|------|
| **Completeness** | | | | | | | | | |
| Overall | 0.001 | 0.001 | 0.001 | 0.001 | 0.001 | 0.001 | 0.001 | 0.001 | 0.001 |
| | (1.00) | (0.74) | (0.64) | (0.76) | (1.00) | (0.62) | (1.00) | (1.00) | (1.00) |
| Summ - Eval | 0.001 | 0.001 | 0.014 | 0.001 | 0.001 | 0.002 | 0.001 | 0.001 | 0.001 |
| | (0.96) | (0.82) | (0.56) | (0.90) | (1.00) | (0.64) | (1.00) | (1.00) | (1.00) |

| Data | Claude 3.7 | Gemn. 2.0 | Gemn. 2.5 | Gemn. 2.5 Pro | GPT - OSS - 20B | GPT-OSS-120B | LL-3.2 | Gemma 3 | Phi - 4 |
|---|---|---|---|---|---|---|---|---|---|
| TL;DR | 0.423 | 0.188 | 0.862 | 0.903 | 0.024 | 0.688 | 0.001 | 0.001 | 0.001 |
| | (0.04) | (0.08) | (-0.30) | (-0.32) | (0.52) | (-0.04) | (1.00) | (1.00) | (1.00) |
| UniSum - Eval | 0.001 | 0.001 | 0.001 | 0.001 | 0.007 | 0.116 | 0.001 | 0.001 | 0.001 |
| | (1.00) | (0.92) | (0.76) | (0.92) | (0.58) | (0.18) | (1.00) | (0.90) | (1.00) |
| **Groundedness** | | | | | | | | | |
| Overall | 0.884 | 0.001 | 0.097 | 0.216 | 0.001 | 0.001 | 0.001 | 0.001 | 0.001 |
| | (-0.36) | (0.96) | (0.90) | (0.66) | (1.00) | (0.38) | (1.00) | (0.74) | (1.00) |
| Summ - Eval | 0.50 | 0.002 | 0.313 | 0.968 | 0.019 | 0.161 | 0.001 | 0.032 | 0.001 |
| | (-0.08) | (0.60) | (0.08) | (-0.10) | (0.58) | (0.26) | (1.00) | (0.38) | (1.00) |
| TL;DR | 0.216 | 0.385 | 0.216 | 0.652 | 0.001 | 0.188 | 0.001 | 0.313 | 0.001 |
| | (0.18) | (-0.04) | (0.20) | (-0.04) | (0.82) | (0.12) | (1.00) | (0.02) | (1.00) |
| UniSum - Eval | 0.862 | 0.138 | 0.615 | 0.976 | 0.688 | 0.014 | 0.001 | 0.053 | 0.001 |
| | (-0.14) | (0.18) | (-0.10) | (-0.28) | (-0.06) | (0.28) | (1.00) | (0.30) | (1.00) |
| Dial - Summ - Eval | 0.001 | 0.001 | 0.097 | 0.215 | 0.001 | 0.001 | 0.001 | 0.001 | 0.001 |
| | (0.98) | (0.92) | (0.04) | (0.10) | (0.82) | (0.66) | (1.00) | (0.88) | (1.00) |
| **Relevance** | | | | | | | | | |
| Overall | 0.001 | 0.001 | 0.001 | 0.001 | 0.001 | 0.001 | 0.001 | 0.001 | 0.001 |
| | (0.92) | (0.84) | (0.88) | (0.88) | (0.92) | (0.68) | (1.00) | (0.80) | (1.00) |
| Dial - Summ - Eval | 0.001 | 0.001 | 0.001 | 0.001 | 0.001 | 0.042 | 0.001 | 0.001 | 0.001 |
| | (1.00) | (0.96) | (0.96) | (1.00) | (0.72) | (0.32) | (1.00) | (0.96) | (1.00) |
| Opin - Summ - Eval | 0.001 | 0.019 | 0.001 | 0.001 | 0.001 | 0.001 | 0.001 | 0.001 | 0.18 |
| | (0.78) | (0.52) | (0.86) | (0.80) | (0.54) | (0.60) | (1.00) | (0.40) | (1.00) |
| Summ - Eval | 0.042 | 0.042 | 0.001 | 0.042 | 0.001 | 0.007 | 0.001 | 0.097 | 0.001 |
| | (0.52) | (0.46) | (0.62) | (0.54) | (0.60) | (0.48) | (0.92) | (0.34) | (1.00) |
| UniSum - Eval | 0.032 | 0.003 | 0.001 | 0.097 | 0.003 | 0.001 | 0.002 | 0.014 | 0.001 |
| | (0.30) | (0.52) | (0.40) | (0.18) | (0.78) | (0.44) | (0.98) | (0.36) | (1.00) |

Table 14: RAG: p-value of Wilcoxon test for the Tau difference between Jury-On-Demand and Static Jury baselines. Numbers in paranthesis () are effect size Cliff's delta.

| Data | Static Jury (Average-All) | Static Jury (Average-TopK) | Static Jury (Weighted-Regression) | Static Jury (Weighted-Tau) |
|---|---|---|---|---|
| **Completeness** | | | | |
| Overall | 0.001 | 0.001 | 0.001 | 0.001 |
| | (1.00) | (1.00) | (1.00) | (1.00) |
| ALCE | 0.001 | 0.001 | 0.001 | 0.001 |
| | (0.44) | (0.84) | (0.80) | (0.94) |
| ASQA | 0.001 | 0.001 | 0.001 | 0.001 |
| | (0.88) | (0.98) | (1.00) | (0.98) |
| QASPER | 0.080 | 0.001 | 0.001 | 0.001 |
| | (0.26) | (0.70) | (0.54) | (0.84) |
| **Groundedness** | | | | |
| Overall | 0.001 | 0.001 | 0.001 | 0.001 |

| Data | Static Jury (Average-All) | Static Jury (Average-TopK) | Static Jury (Weighted-Regression) | Static Jury (Weighted-Tau) |
|---|---|---|---|---|
| | (1.00) | (1.00) | (0.88) | (1.00) |
| CAQA | 0.001 | 0.001 | 0.001 | 0.001 |
| | (0.98) | (1.00) | (0.92) | (1.00) |
| Halu - Eval | 0.001 | 0.003 | 0.001 | 0.001 |
| | (0.94) | (0.70) | (0.90) | (1.00) |
| Rag - Truth | 0.001 | 0.001 | 0.005 | 0.001 |
| | (0.54) | (1.00) | (0.56) | (1.00) |
| **Relevance** | | | | |
| Overall | 0.001 | 0.001 | 0.042 | 0.001 |
| | (0.80) | (1.00) | (0.44) | (0.84) |
| ALCE | 0.001 | 0.001 | 0.216 | 0.001 |
| | (0.16) | (1.00) | (0.12) | (1.00) |
| Hotpot - QA | 0.001 | 0.001 | 0.001 | 0.001 |
| | (0.74) | (0.88) | (0.96) | (0.98) |
| MS MARCO | 0.001 | 0.722 | 0.161 | 0.754 |
| | (0.50) | (-0.06) | (0.08) | (0.04) |

Table 15: RAG: p-value of Wilcoxon test for the Tau difference between Jury-On-Demand and single judge. Numbers in paranthesis () are effect size Cliff's delta.

| Data | Claude 3.7 | Gemn. 2.0 | Gemn. 2.5 | Gemn. 2.5 Pro | GPT - OSS - 20B | GPT - OSS - 120B | LL - 3.2 | Gemma 3 | Phi - 4 |
|---|---|---|---|---|---|---|---|---|---|
| **Completeness** | | | | | | | | | |
| Overall | 0.001 | 0.001 | 0.001 | 0.001 | 0.001 | 0.001 | 0.001 | 0.001 | 0.001 |
| | (1.00) | (1.00) | (1.00) | (1.00) | (1.00) | (1.00) | (1.00) | (1.00) | (1.00) |
| ALCE | 0.002 | 0.003 | 0.001 | 0.001 | 0.042 | 0.002 | 0.001 | 0.001 | 0.001 |
| | (0.48) | (0.62) | (0.82) | (0.80) | (0.24) | (0.50) | (1.00) | (0.78) | (1.00) |
| ASQA | 0.001 | 0.001 | 0.001 | 0.001 | 0.001 | 0.001 | 0.001 | 0.001 | 0.001 |
| | (0.84) | (0.94) | (1.00) | (1.00) | (1.00) | (0.96) | (1.00) | (0.98) | (1.00) |
| QASPER | 0.002 | 0.001 | 0.216 | 0.042 | 0.032 | 0.50 | 0.001 | 0.002 | 0.001 |
| | (0.62) | (0.84) | (0.10) | (0.44) | (0.20) | (-0.04) | (1.00) | (0.5) | (1.00) |
| **Groundedness** | | | | | | | | | |
| Overall | 0.001 | 0.001 | 0.001 | 0.001 | 0.001 | 0.001 | 0.001 | 0.001 | 0.001 |
| | (1.00) | (1.00) | (1.00) | (1.00) | (1.00) | (0.98) | (1.00) | (1.00) | (1.00) |
| CAQA | 0.001 | 0.001 | 0.001 | 0.001 | 0.001 | 0.001 | 0.001 | 0.001 | 0.001 |
| | (1.00) | (1.00) | (0.98) | (1.00) | (1.00) | (0.98) | (1.00) | (1.00) | (1.00) |
| Halu - Eval | 0.001 | 0.001 | 0.001 | 0.001 | 0.019 | 0.042 | 0.001 | 0.001 | 0.001 |
| | (1.00) | (1.00) | (0.90) | (0.88) | (0.48) | (0.12) | (1.00) | (1.00) | (1.00) |
| Rag - Truth | 0.001 | 0.001 | 0.001 | 0.001 | 0.001 | 0.001 | 0.001 | 0.001 | 0.001 |
| | (1.00) | (1.00) | (0.28) | (0.80) | (0.76) | (0.66) | (1.00) | (1.00) | (1.00) |
| **Relevance** | | | | | | | | | |
| Overall | 0.001 | 0.001 | 0.001 | 0.001 | 0.001 | 0.001 | 0.001 | 0.001 | 0.001 |
| | (1.00) | (1.00) | (0.90) | (0.84) | (0.90) | (0.80) | (1.00) | (1.00) | (1.00) |
| ALCE | 0.001 | 0.001 | 0.001 | 0.001 | 0.001 | 0.001 | 0.001 | 0.001 | 0.001 |
| | (0.92) | (0.84) | (0.48) | (0.38) | (0.24) | (0.34) | (1.00) | (1.00) | (1.00) |
| Hotpot - QA | 0.001 | 0.001 | 0.001 | 0.001 | 0.001 | 0.001 | 0.001 | 0.001 | 0.001 |
| | (0.84) | (0.94) | (0.44) | (0.48) | (0.50) | (0.36) | (1.00) | (1.00) | (1.00) |

| Data | Claude 3.7 | Gemn. 2.0 | Gemn. 2.5 | Gemn. 2.5 Pro | GPT - OSS - 20B | GPT - OSS - 120B | LL - 3.2 | Gemma 3 | Phi - 4 |
|---|---|---|---|---|---|---|---|---|---|
| MS MARCO | 0.001 | 0.001 | 0.001 | 0.001 | 0.001 | 0.001 | 0.001 | 0.001 | 0.001 |
| | (0.58) | (0.70) | (0.38) | (0.26) | (0.80) | (0.36) | (1.00) | (1.00) | (1.00) |

## E.2 ANALYZING THE INTERACTION BETWEEN JUDGES, TASKS, AND DATA ATTRIBUTES

Fig. 5 summarizes the selection frequency of each judge, revealing that RAG juries tend to include a broader set of judges compared to those used in summarization tasks. Within summarization, Claude 3.7 Sonnet is frequently selected for groundedness evaluation, but is rarely chosen for completeness. Conversely, Gemini 2.5 Flash and Gemini 2.0 Flash are often selected for completeness, yet seldom appear in groundedness evaluations.

Fig. 6 details judge selection frequencies for individual datasets within the RAG groundedness task. The jury for this task comprises eight judges, and for clarity, Fig. 6 highlights the top three selected judges. Notably, GPT-OSS-120B is more frequently ranked as the top-1 or top-2 judge for HaluEval data. Gemini 2.5 Flash dominates the top selections for RagTruth data, but is rarely selected for HaluEval. Claude 2.7 Sonnet is also selected more frequently for RagTruth compared to HaluEval and CAQA.

Fig. 7 shows the judge selection frequencies for datasets within the summarization completeness task. We observe that Gemini 2.0 Flash is selected more frequently than Gemini 2.5 Flash for SummEval and OpinSumm, while the opposite is true for UniSumm. Additionally, GPT-OSS-20B is chosen far more often in UniSumm compared to SummEval and OpinSumm.

To further explore how data properties influence judge selection, we examine selection patterns across different bins of key attributes, using CAQA from the RAG groundedness task as a case study. The jury for this task comprises eight judges, and for clarity, Figure 8 highlights the top three selected judges. The results reveal distinct trends: Compared to Gemini models, GPT-OSS-120B and GPT-OSS-20B are more frequently selected among the top-3 judges when the source context compression ratio is low (i.e., below 0.75), whereas Claude 3.7 Sonnet is more often selected when the compression ratio is high. Additionally, Llama and Phi models also show increased selection frequency under high compression ratios. For response token entropy, Claude 3.7 Sonnet and Gemini 2.0 Flash show higher selection frequency than Gemini 2.5 Flash and Gemini 2.5 Pro when the response token entropy is low (below 3.6), while Gemini 2.5 Flash and Gemini 2.5 Pro are selected more frequently when the response token entropy is high.

These findings reinforce the importance of constructing dynamic juries that adapt to specific data characteristics, and demonstrate the potential of predicting judge reliability based on interpretable data properties.

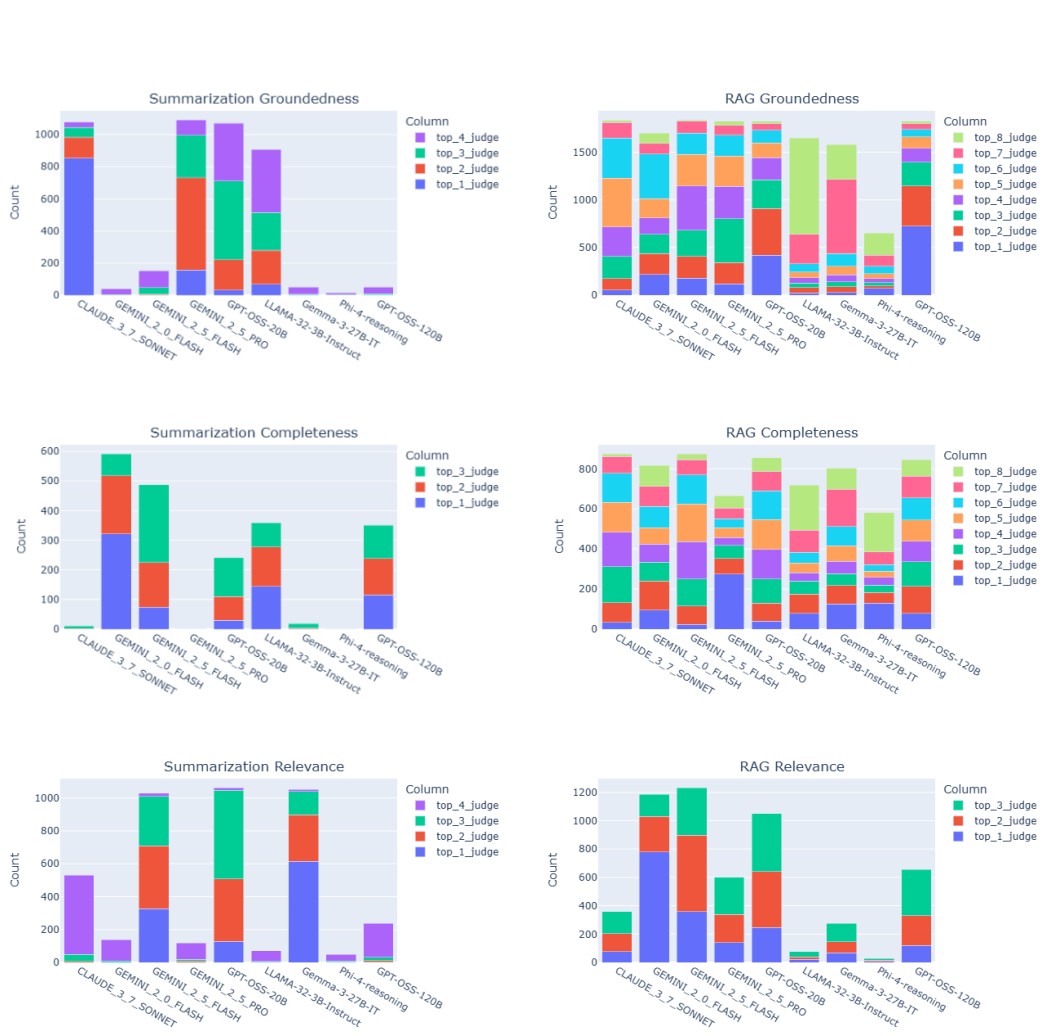

Figure 5: Selection frequency of the judge in the jury. Top k judge means that the judge has the k-th highest reliability score in the jury. RAG juries tend to incorporate a more diverse set of judges compared to those used in summarization tasks. For summarization, Claude 3.7 Sonnet is frequently chosen for groundedness assessments but is rarely selected for completeness. In contrast, Gemini 2.5 Flash and Gemini 2.0 Flash are commonly selected for completeness, yet seldom appear in groundedness evaluations.

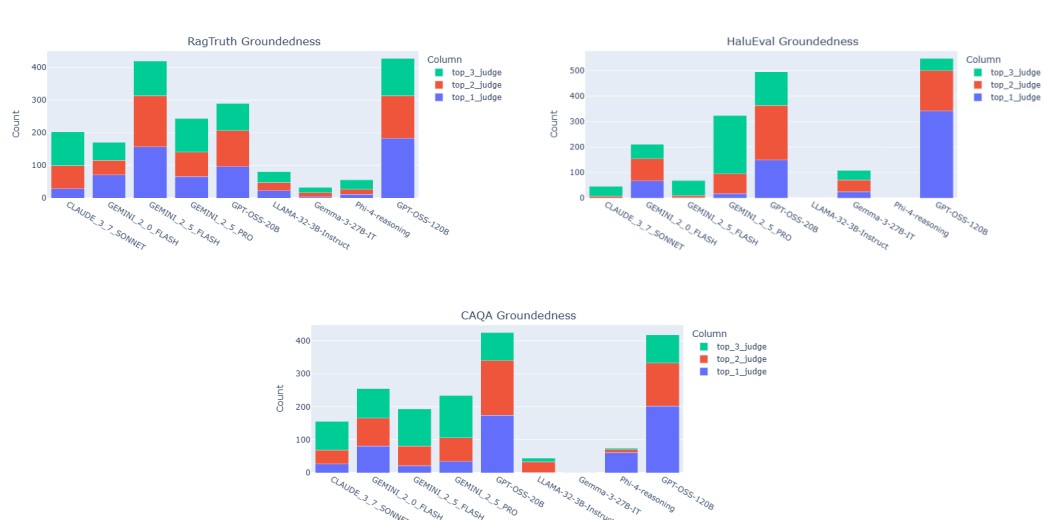

Figure 6: Judge selection in the jury for RAG groundedness. GPT-OSS-120B is more frequently ranked as the top-1 or top-2 judge for HaluEval data. Gemini 2.5 Flash dominates the top selections for RagTruth data, but is rarely selected for HaluEval. Claude 2.7 Sonnet is also selected more frequently for RagTruth compared to HaluEval and CAQA.

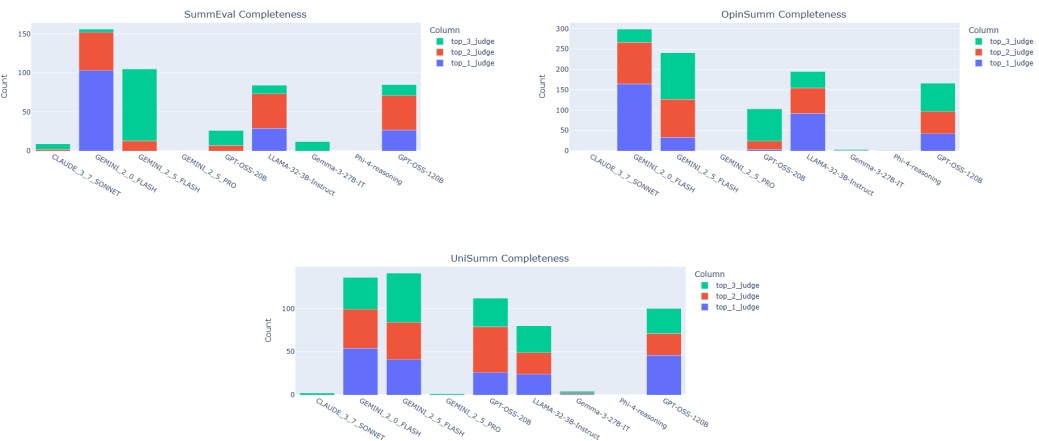

Figure 7: Judge selection in the jury for summarization completeness. Gemini 2.0 Flash is selected more frequently than Gemini 2.5 Flash for SummEval and OpinSumm, while the opposite is true for UniSumm. Additionally, GPT-OSS-20B is chosen far more often in UniSumm compared to SummEval and OpinSumm.

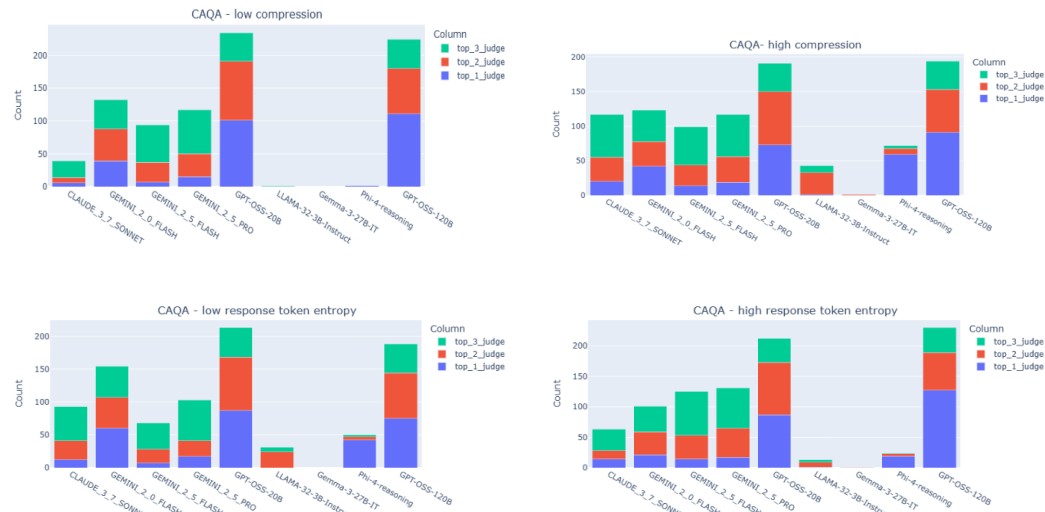

Figure 8: Variation in judge selection for RAG groundedness (CAQA) across compression ratio and response token entropy. GPT-OSS-120B and GPT-OSS-20B are more frequently selected among the top-3 judges when the source context compression ratio is low (i.e., below 0.75), whereas Claude 3.7 Sonnet is more often selected when the compression ratio is high. For response token entropy, Claude 3.7 Sonnet and Gemini 2.0 Flash show higher selection frequency than Gemini 2.5 Flash and Gemini 2.5 Pro when the response token entropy is low (below 3.6), while Gemini 2.5 Flash and Gemini 2.5 Pro are selected more frequently when the response token entropy is high.

Tables 16-17 present the top five most important features for each judge's XGBoost model, as determined by permutation feature importance (Fisher et al., 2019), using the summarization groundedness task as an illustrative example. The results show substantial variation in the top-ranked features across different judges, suggesting that each judge's reliability is influenced by distinct data properties.

Fig. 9 aggregates the top five features that frequently appear across tasks, revealing clear task-specific patterns. For instance, compression ratio and token entropy are more prominent in RAG tasks, while embedding-related features such as PCA components and embedding similarity are more influential in summarization tasks. These findings align with the ablation analysis in Appendix H.1, which shows that removing embedding features leads to a greater performance drop in the summarization task compared to RAG. For RAG, removing text size-related features results in a larger decline than removing embedding features.

These observations imply that evaluation reliability is task-dependent, and further demonstrate that our approach effectively links data characteristics to judge reliability, enabling more informed and adaptive jury construction across diverse evaluation scenarios.

Table 16: The top 3 most important features for each judge's XGBoost model from summarization groundedness. The results show substantial variation in the top-ranked features across different judges.

| Judge | Feature 1 | Feature 2 | Feature 3 |
|---|---|---|---|
| CLAUDE 3 .7 SONNET | input _ LEXICAL _ DIVERSITY | input _ COREFERENCE _ CHAIN | output _pca _6 |
| GEMINI 2.0 FLASH | output_pca_8 | output _SEMANTIC _AMBIGUITY | output _pca _1 |
| GEMINI 2.5 FLASH | output _READING _INDEX | output _pca _8 | output _NUM _WORD _SENTENCE |
| GEMINI 2.5 PRO | output _pca _2 | input _pca _1 | input _NEGATION _SENTENCE |
| GPT-OSS-20B | output _embedding _similarity _education | output _embedding _similarity _business | output _NUM _CHAR _WORD |
| LLAMA-32-3B-Instruct | output _STOP _WORDS | output _CHAR _COMPRESSION | output _NGRAM _COUNT |
| Gemma-3-27B-IT | input _TOKEN _ENTROPY | input _DIFFICULT _WORD | input _pca _2 |
| Phi-4-reasoning | input _DIFFICULT _WORD | output _pca _5 | output _pca _1 |
| GPT-OSS-120B | output _READING _INDEX | input _embedding _similarity _business | output _NGRAM _REPETITION |

Table 17: The 4th and 5th most important features for each judge's XGBoost model from summarization groundedness. The results show substantial variation in the top-ranked features across different judges.

| Judge | Feature 4 | Feature 5 |
|---|---|---|
| CLAUDE 3 .7 SONNET | input _SEMANTIC _AMBIGUITY | output _embedding _similarity _business |
| GEMINI 2.0 FLASH | input _NUMBER _COUNT | output _pca _7 |
| GEMINI 2.5 FLASH | output _pca _7 | input _embedding _similarity _science |
| GEMINI 2.5 PRO | input _pca _9 | output _embedding _similarity _education |
| GPT-OSS-20B | output _TOKEN _ENTROPY | input _pca _1 |
| LLAMA-32-3B-Instruct | output _LEXICAL _DIVERSITY | output _NUMBER _COUNT |
| Gemma-3-27B-IT | output _FACTUAL _DENSISTY | input _FACTUAL _DENSISTY |
| Phi-4-reasoning | input _NGRAM _COUNT | output _SUBJECTIVITY |
| GPT-OSS-120B | output _embedding _similarity _business | input _MODALITY |

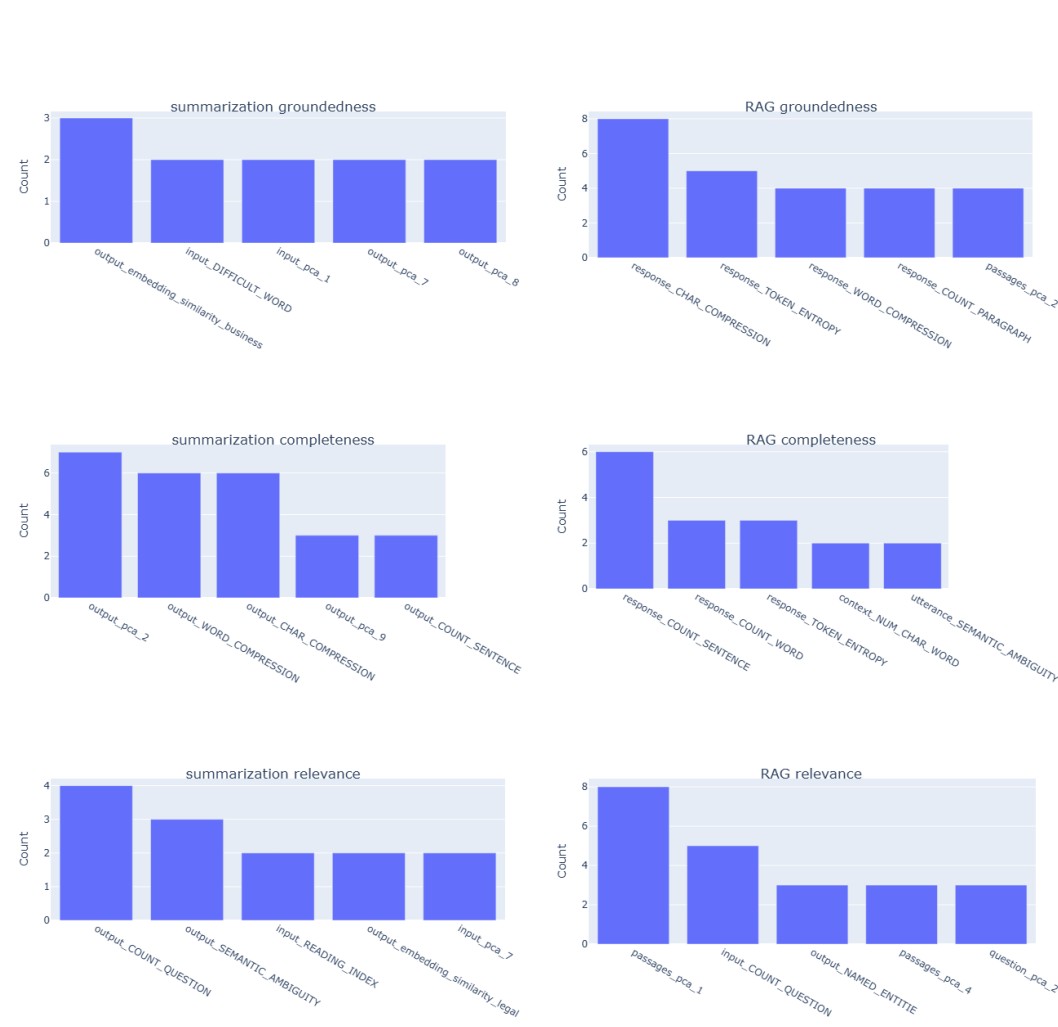

Figure 9: Aggregating the top five features that frequently appear across tasks. Compression ratio and token entropy are more prominent in RAG tasks, while embedding-related features such as PCA components and embedding similarity are more influential in summarization tasks.

## F   Human Evaluation Consistency

We examine the consistency of human evaluations and compare them with the performance of our jury model. Both the SummEval and DialSumm datasets include three human annotators per evaluation dimension. Tables 18 to 21 report Kendall's tau correlations between each individual annotation and the average of the three scores, which serves as the reference in our analysis. The top-left cell in each table presents the Kendall's tau of the jury model. Our findings reveal that inter-annotator agreement is generally low, highlighting notable discrepancies among human judgments. While Kendall's tau increases when comparing individual annotator scores to the average, substantial variation remains. For instance, in the SummEval Relevance dimension, the highest tau between a human annotator and the average is 0.875, whereas the lowest is only 0.411. Although the jury model rarely surpasses the best-performing annotator, it consistently outperforms the weaker ones. This suggests that the jury model offers a reliable and robust alternative to individual human evaluations.

Table 18: Kendall's tau for human annotations – SummEval Groundedness

| 0.576 | Average score | Annotator 1 | Annotator 2 | Annotator 3 |
|---|---|---|---|---|
| **Average score** | 1 | 0.893 | 0.893 | 0.735 |
| **Annotator 1** | 0.893 | 1 | 0.748 | 0.793 |
| **Annotator 2** | 0.893 | 0.748 | 1 | 0.807 |
| **Annotator 3** | 0.735 | 0.793 | 0.807 | 1 |

Table 19: Kendall's tau for human annotations – SummEval Relevance.

| 0.696 | Average score | Annotator 1 | Annotator 2 | Annotator 3 |
|---|---|---|---|---|
| **Average score** | 1 | 0.667 | 0.875 | 0.411 |
| **Annotator 1** | 0.667 | 1 | 0.455 | 0.388 |
| **Annotator 2** | 0.875 | 0.455 | 1 | 0.394 |
| **Annotator 3** | 0.411 | 0.388 | 0.394 | 1 |

Table 20: Kendall's tau for human annotations – DialSumm Groundedness.

| 0.699 | Average score | Annotator 1 | Annotator 2 | Annotator 3 |
|---|---|---|---|---|
| **Average score** | 1 | 0.647 | 0.712 | 0.679 |
| **Annotator 1** | 0.647 | 1 | 0.462 | 0.379 |
| **Annotator 2** | 0.712 | 0.463 | 1 | 0.351 |
| **Annotator 3** | 0.679 | 0.379 | 0.351 | 1 |

Table 21: Kendall's tau for human annotations – DialSumm Relevance.

| 0.639 | Average score | Annotator 1 | Annotator 2 | Annotator 3 |
|---|---|---|---|---|
| **Average score** | 1 | 0.741 | 0.758 | 0.622 |
| **Annotator 1** | 0.741 | 1 | 0.564 | 0.365 |
| **Annotator 2** | 0.758 | 0.564 | 1 | 0.344 |
| **Annotator 3** | 0.622 | 0.365 | 0.344 | 1 |

## G   Judge Reliability Prediction Model Performance

Table 22 presents the AUC scores of ROC curves for each judge reliability model trained using XGBoost on the testing set. The AUC values are relatively consistent across tasks; we

include results for summarization completeness and RAG groundedness as representative examples. Result varies across judges and tasks, with most AUCs ranging between 0.66 and 0.76, indicating that the models demonstrate adequate predictive capability for these evaluation tasks.

Table 22: AUC of ROC for the judge reliability model. Here Gemn. is Gemini.

| Metric | Claude 3.7 SON-NET | Gemn. 2.0 Flash | Gemn. 2.5 Flash | Gemn. 2.5 Pro | GPT-OSS-20B | GPT-OSS-120B | LLAMA -3.2-3B Instruct | Gemma 3-27B-IT | Phi-4-reason-ing |
|---|---|---|---|---|---|---|---|---|---|
| Summarization-completeness | 0.66 | 0.68 | 0.69 | 0.66 | 0.72 | 0.70 | 0.67 | 0.76 | 0.73 |
| RAG-groundedness | 0.65 | 0.75 | 0.66 | 0.69 | 0.71 | 0.63 | 0.75 | 0.76 | 0.70 |

# H ABLATION STUDY AND MODEL WEAKNESS ANALYSIS

## H.1 ABLATION STUDY (DATA PROPERTIES)

To investigate the influence of data property features on jury performance, we conduct ablation studies by selectively removing feature sets during model construction. As illustrative examples, we focus on two tasks: Summarization-Completeness and RAG-Groundedness. The full categorization of features is provided in Appendix A. Specifically, we remove three groups of features in separate experiments: (1) text size-related features and special word count features (jointly removed due to their high correlation), (2) text complexity features, and (3) embedding-based features. The results are presented in Tables 23 and 24 for Summarization-Completeness and RAG-Groundedness, respectively. We observe that the jury model achieves its best performance when all feature sets are included, underscoring the importance of comprehensive feature representation. Although the performance differences are modest, this can be attributed to the internal correlations within each feature category. Additionally, different tasks exhibit varying sensitivity to feature sets. For instance, removing embedding features leads to a greater performance drop in the Summarization-Completeness task than in RAG-Groundedness, where text size-related features have a more pronounced impact.

Table 23: Kendall's tau for jury performance on Summarization-Completeness under feature ablation. The number of judges ($K$) is indicated for each configuration. The jury using all features achieves the highest performance, particularly on SummEval.

| Data | Text size + special words removed ($K = 3$) | Text complexity removed ($K = 3$) | Embedding removed ($K = 4$) | All features ($K = 3$) |
|---|---|---|---|---|
| Overall | 0.444 | 0.469 | 0.459 | 0.494 |
| SummEval | 0.633 | 0.563 | 0.545 | 0.696 |
| TL;DR | 0.388 | 0.379 | 0.363 | 0.396 |
| UniSumEval | 0.628 | 0.593 | 0.614 | 0.609 |

Table 24: Kendall's tau for jury performance on RAG-Groundedness under feature ablation. The number of judges ($K$) is indicated for each configuration. The jury using all features achieves the highest performance, particularly on RagTruth.

| Data | Text size + special words removed ($K = 4$) | Text complexity removed ($K = 7$) | Embedding removed ($K = 8$) | All features ($K = 7$) |
|------|------|------|------|------|
| Overall | 0.671 | 0.679 | 0.686 | 0.698 |
| CAQA | 0.714 | 0.729 | 0.734 | 0.751 |
| HaluEval | 0.757 | 0.720 | 0.721 | 0.744 |
| RagTruth | 0.498 | 0.498 | 0.475 | 0.525 |

## H.2 Ablation Study (Jury Size - $K$)

We conduct experiments to test the effectiveness of varying jury size compared to keeping a fixed value. We focus on the tasks Summarization-Completeness and RAG-Groundedness for this study. The tolerance level for all trained XGBoost models is set to 0 which means only the exactly matching scores are considered as correct. Performance is measured across 10 runs and average is taken for each jury size. Fig. 10 shows that the performance varies with jury size. In the Summarization-Completeness task it gives the best performance with a jury size of 3. Performance first increases with increasing jury size as better judges together provide a better overall decision. It then decreases as poor judges need to be added to the jury, impacting the overall jury decision. In the RAG-Groundedness task performance is poor at lower jury sizes and improves with bigger jury sizes. In this case, adding more judges benefits the overall jury decision by making it more robust against noisy predictions. This shows us that tuning on jury size gives significant improvements than keeping a fixed one from task to task.

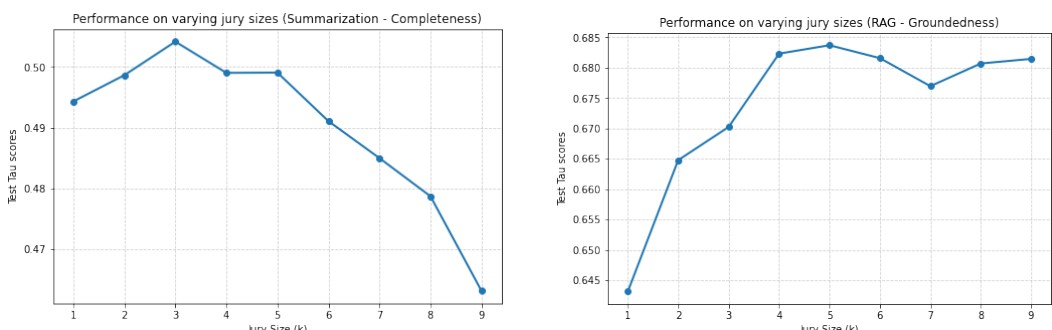

Figure 10: Test performance with varying jury size.

## H.3 Ablation Study (Tolerance levels - $\tau$)

We consider two tolerance levels for summarization tasks (0 and 1) and a single tolerance level (0) for RAG tasks. For RAG tasks valid scores are 0-2. Thus, a tolerance level of 1 or more would mean that a score of 1 is always considered as correct. For the same reason, we do not experiment with a tolerance level of 2 or more for summarization tasks where the valid scores are 1-5.

For this study, we focus on Summarization-Completeness. We experiment with two tolerance levels: 0 and 1 in the original scale (1-5). After min-max normalization to [0, 1], these correspond to 0 and 0.25. The performance of the jury is observed with all the XGBoost models trained either on tolerance of 0 or 0.25. Table 25 summarizes the means and standard deviations of the Kendall's Tau across the 10 runs and shows the comparison with tuned tolerance models. We observe that allowing different tolerance levels across different XGBoost models gives slightly better performance than a fixed tolerance level across all models.

Additionally, Table 25 further illustrates the importance of tolerance tuning. While Jury-on-Demand with variable tolerance achieves the best overall performance, the optimal fixed tolerance differs across datasets: TL;DR performs best with a tolerance of 0, whereas UniSumEval favors 0.25. This variability underscores that no single fixed tolerance can fit all datasets. In practical scenarios, especially for unseen datasets without human annotations, it is impossible to know the ideal tolerance beforehand. Therefore, adaptive approaches that allow tolerance to vary across models or instances are crucial for robust generalization.

Finally, the results with fixed tolerance levels are better than the static jury as we have chosen the best jury size ($K = 3$) overall across the runs.

Table 25: Kendall's tau for jury performance on Summarization-Completeness under tolerance ablation. The number of judges ($K$) is fixed to 3 (overall best) for the fixed tolerance configurations.

| Data | Fixed tolerance (0) | Fixed tolerance (0.25) | Jury on Demand (variable tolerance) | Static Jury |
|---|---|---|---|---|
| Overall | 0.49 (0.01) | 0.50 (0.01) | 0.51 (0.03) | 0.46 (0.02) |
| SummEval | 0.70 (0.06) | 0.70 (0.03) | 0.74 (0.05) | 0.62 (0.05) |
| TL;DR | 0.46 (0.04) | 0.50 (0.03) | 0.39 (0.06) | 0.39 (0.03) |
| UniSumEval | 0.71 (0.04) | 0.66 (0.04) | 0.68 (0.05) | 0.61 (0.05) |

### H.4 Model Weakness Analysis

To identify conditions where the jury model underperforms, we analyzed feature distributions between instances with high and low prediction discrepancies - excluding embedding features for interpretability. We focus on the RAG-Groundedness task for illustration. The dataset was split into two groups:

1. High Difference: Instances with large prediction errors.

2. Low/No Difference: Instances with minimal errors.

For each feature, we computed mean and median values across both groups and ranked features by their median absolute differences in Table 26. This revealed two failure modes:

1. **Systematic Bias:** Features like factual density and named entities show consistent shifts in both mean and median, suggesting bias toward certain content structures.

2. **Distributional Fragility:** Features such as syntactic anomaly and subjectivity show high median shifts but low mean differences, indicating sensitivity to rare or irregular linguistic patterns.

Table 26: Top 5 features ranked by median differences between high-difference and low/no-difference groups.

| Feature | Median Difference | Mean Difference |
|---|---|---|
| utterance_SYNTACTIC_ANOMALY | 1.97 | 0.11 |
| utterance_NAMED_ENTITY | 0.77 | 0.37 |
| utterance_SYNTACTIC_AMBIGUITY | 0.74 | 0.22 |
| context_FACTUAL_DENSISTY | 0.63 | 0.36 |
| context_SUBJECTIVITY | 0.53 | 0.17 |

# I   FRAMEWORK GENERALIZABILITY TO UNSEEN DOMAINS

To assess the framework's generalizability to unseen domains, we employ a leave-one-out procedure. Specifically, for each experiment, one data source is excluded from the training of XGBoost reliability models and jury construction, and the trained framework is then evaluated on the held-out source. This approach tests whether the Jury-on-Demand mechanism consistently outperforms both static juries and individual judges in previously unseen domains.

Using the RAG-Relevance task as an illustrative example, the dataset comprises three sources: ALCE, Hotpot-QA, and MS MARCO. In the first iteration, ALCE is held out while the framework is trained on Hotpot-QA and MS MARCO; performance is then assessed on ALCE. The process is repeated for each remaining source. The complete results are presented in Table 27. Across all three cases, Jury-on-Demand achieves the highest performance, indicating that the learned patterns generalize effectively to held-out domains.

Table 27: Kendall's tau on held-out data source - RAG Relevance

| Held-out | Jury-on-Demand | Static-Jury | Claude 3.7 | Gemn. 2.0 | Gemn. 2.5 | Gemn. 2.5 Pro | GPT-OSS-20B | GPT-OSS-120B | LL-3.2 | Gemma 3 | Phi-4 |
|---|---|---|---|---|---|---|---|---|---|---|---|
| ALCE | 0.59 | 0.58 | 0.52 | 0.51 | 0.53 | 0.55 | 0.57 | 0.58 | 0.11 | 0.43 | 0.12 |
| Hotpot-QA | 0.89 | 0.87 | 0.85 | 0.84 | 0.88 | 0.88 | 0.88 | 0.88 | 0.40 | 0.77 | 0.46 |
| MS-MARCO | 0.46 | 0.42 | 0.42 | 0.39 | 0.43 | 0.43 | 0.43 | 0.43 | 0.14 | 0.17 | 0.12 |

We extend our evaluation to the Summarization-Groundedness task, which includes four data sources: Summ-Eval, TL;DR, UniSum-Eval, and Dial-Summ-Eval. Table 28 reports the performance of the Jury-on-Demand framework under the leave-one-out setting for each source. The results indicate strong generalization for three sources, while performance on Dial-Summ-Eval is weaker.

These findings reinforce that the framework's generalizability is influenced by both the diversity of training data and the characteristics of unseen domains. As additional annotated datasets become available and incorporated into training, we expect the framework's ability to generalize to new domains to improve substantially.

Table 28: Kendall's tau on held-out data source - Summarization Groundedness

| Held-out | Jury-on-Demand | Static-Jury | Claude 3.7 | Gemn. 2.0 | Gemn. 2.5 | Gemn. 2.5 Pro | GPT-OSS-20B | GPT-OSS-120B | LL-3.2 | Gemma 3 | Phi-4 |
|---|---|---|---|---|---|---|---|---|---|---|---|
| Dial-Summ-Eval | 0.63 | 0.67 | 0.61 | 0.64 | 0.71 | 0.70 | 0.65 | 0.68 | 0.28 | 0.66 | 0.27 |
| Summ-Eval | 0.64 | 0.62 | 0.60 | 0.55 | 0.62 | 0.62 | 0.54 | 0.60 | 0.20 | 0.57 | 0.14 |
| TL;DR | 0.43 | 0.43 | 0.36 | 0.40 | 0.38 | 0.43 | 0.27 | 0.40 | 0.08 | 0.38 | 0.07 |
| Uni-Summ-Eval | 0.68 | 0.65 | 0.61 | 0.59 | 0.63 | 0.64 | 0.63 | 0.63 | 0.17 | 0.58 | 0.35 |

## J  Judge Score Calibration

Some judges may consistently assign lower or higher scores compared to human annotations. To address this, we apply score calibration. Specifically, we perform isotonic calibration (Niculescu-Mizil & Caruana, 2005) for each judge's score within each dataset, then retrain the XGBoost model and construct the jury using the calibrated scores. For illustration, we focus on summarization completeness and groundedness. Prior to calibration, we examine the difference between each judge's mean score and the mean human annotation score (annotation score minus judge score). Fig. 11 show these differences for completeness and groundedness in Unisumm. We observe that weaker models—particularly LLAMA 3.2 3B Instruct, Gemma 3.2 7B IT, and Phi 4 Reasoning—tend to exhibit larger discrepancies. Additionally, nearly all judges assign lower scores than human annotations for groundedness in Unisumm, suggesting a potential need for calibration.

Tables 30 and 29 compare Kendall's tau before and after calibration. Overall performance changes are minimal, with calibrated results slightly improving on groundedness (0.63 vs. 0.612) but slightly worsening on completeness (0.476 vs. 0.49). Differences become more pronounced within certain datasets and for specific judges. For example, as shown in Fig. 12 (left), tau for Gemini 2.5 Flash on Unisumm completeness drops from 0.592 to 0.53 because many original score 5s are calibrated to 4, leading to underestimation of human annotation scores. Conversely, for Unisumm groundedness (Fig. 12 (right)), Gemini 2.5 Flash's tau increases from 0.6 to 0.65 after calibration because scores of 2 and 3 are calibrated to 4, reducing bias in judge scores. In summary, calibration can be beneficial for certain tasks and judges, but it may also introduce under- or overestimation of human annotations, reducing alignment. Future work will explore strategies to mitigate judge score bias more effectively.

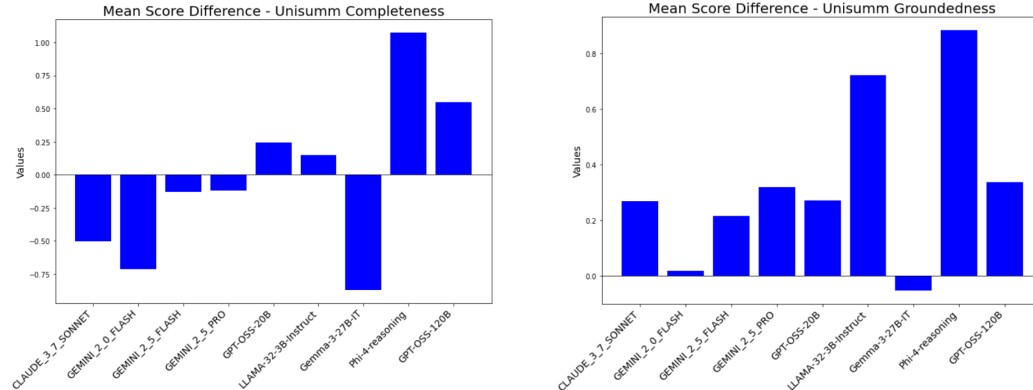

Figure 11: The difference between each judge's mean score and the mean human annotation score (annotation score minus judge score) for completeness and groundedness in Unisumm.

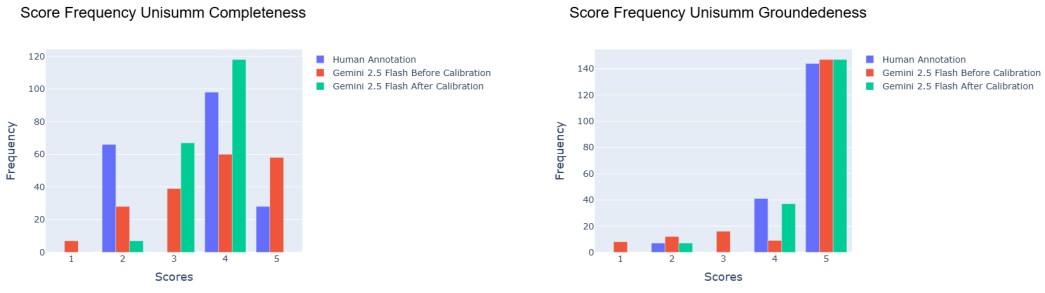

Figure 12: The difference between each judge's mean score and the mean human annotation score (annotation score minus judge score) for completeness and groundedness in Unisumm.

Table 29: Summarization completeness: Kendall's tau after calibration, with original values shown in parentheses. N/A indicates cases where all scores were calibrated to a single value, making Kendall's tau computation infeasible.

| Model | Jury-on-Demand | Claude 3.7 | Gemn. 2.0 | Gemn. 2.5 | Gemn. 2.5 Pro | GPT-OSS-20B | GPT-OSS-120B | LL-3.2 | Gemma 3 | Phi-4 |
|---|---|---|---|---|---|---|---|---|---|---|
| Overall | 0.476 | 0.376 | 0.450 | 0.477 | 0.411 | 0.370 | 0.443 | 0.196 | 0.256 | 0.202 |
| | (0.490) | (0.395) | (0.415) | (0.444) | (0.396) | (0.323) | (0.448) | (0.105) | (0.264) | (0.184) |
| Summ-Eval | 0.627 | 0.568 | 0.593 | 0.638 | 0.492 | N/A | 0.624 | N/A | 0.075 | 0.244 |
| | (0.723) | (0.568) | (0.583) | (0.638) | (0.497) | (0.035) | (0.624) | (0.053) | (0.161) | (0.239) |
| TL;DR | 0.400 | 0.344 | 0.430 | 0.429 | 0.392 | 0.346 | 0.393 | 0.077 | 0.184 | N/A |
| | (0.441) | (0.344) | (0.391) | (0.429) | (0.392) | (0.346) | (0.393) | (0.130) | (0.184) | (0.110) |
| Uni-Summ-Eval | 0.673 | 0.441 | 0.513 | 0.530 | 0.460 | 0.659 | 0.697 | 0.435 | 0.607 | 0.361 |
| | (0.725) | (0.415) | (0.511) | (0.592) | (0.535) | (0.649) | (0.695) | (0.435) | (0.597) | (0.258) |

Table 30: Summarization groundedness: Kendall's tau after calibration, with original values shown in parentheses. N/A indicates cases where all scores were calibrated to a single value, making Kendall's tau computation infeasible.

| Model | Jury-on-Demand | Claude 3.7 | Gemn. 2.0 | Gemn. 2.5 | Gemn. 2.5 Pro | GPT-OSS-20B | GPT-OSS-120B | LL-3.2 | Gemma 3 | Phi-4 |
|---|---|---|---|---|---|---|---|---|---|---|
| Overall | 0.629 | 0.618 | 0.573 | 0.626 | 0.542 | 0.603 | 0.581 | 0.360 | 0.641 | 0.417 |
| | (0.612) | (0.573) | (0.498) | (0.507) | (0.515) | (0.480) | (0.571) | (0.244) | (0.561) | (0.239) |
| Dial-Summ-Eval | 0.683 | 0.623 | 0.579 | 0.722 | 0.656 | 0.710 | 0.687 | 0.260 | 0.686 | 0.286 |
| | (0.727) | (0.630) | (0.624) | (0.718) | (0.656) | (0.673) | (0.687) | (0.278) | (0.700) | (0.262) |
| Summ-Eval | 0.698 | 0.622 | 0.549 | 0.694 | 0.661 | 0.649 | 0.684 | 0.286 | 0.631 | N/A |
| | (0.660) | (0.628) | (0.550) | (0.686) | (0.663) | (0.651) | (0.687) | (0.289) | (0.631) | (0.232) |
| TL;DR | 0.388 | 0.355 | 0.460 | 0.365 | 0.353 | 0.277 | 0.464 | 0.015 | 0.489 | 0.114 |
| | (0.494) | (0.391) | (0.458) | (0.363) | (0.375) | (0.284) | (0.464) | (0.015) | (0.462) | (0.070) |
| Uni-Summ-Eval | 0.614 | 0.509 | 0.584 | 0.650 | 0.584 | 0.649 | 0.582 | 0.332 | 0.571 | 0.335 |
| | (0.582) | (0.540) | (0.584) | (0.600) | (0.605) | (0.619) | (0.583) | (0.233) | (0.596) | (0.162) |

## K    PARAMETER TUNING IN XGBOOST

We use random search to tune the hyperparameters of the XGBoost models for judge reliability scores. The search space for the tuned parameters is provided in Table 31. Parameters not listed in the table are set to their default values.

Table 31: Parameter search space in XGBoost.

| Parameter | Search Space |
|---|---|
| max_depth | 2, 3, 4, 5, 6, 7, 8, 9 |
| learning_rate | 0.01, 0.03, 0.05, 0.07, 0.1, 0.2 |
| n_estimators | 500, 600, 800, 1000, 1200 |

