# OpenReview forum: "Who Judges the Judge? LLM Jury-on-Demand: Building Trustworthy LLM Evaluation Systems"
_ICLR.cc/2026/Conference — Submitted to ICLR 2026_

### Official Review · Reviewer_3GyE · 2025-10-18

**Soundness:** 2
**Presentation:** 3
**Contribution:** 2
**Rating:** 4
**Confidence:** 4

**Summary:**

This paper proposes "LLM Jury-on-Demand," a dynamic, learning-based framework for evaluating LLM outputs. The core idea is to move beyond static evaluation methods by training a set of "reliability predictors" that assess, on an model-by-model basis, how likely each LLM judge is to agree with human experts. This allows the system to dynamically assemble an optimal jury for each data point and assign instance-specific weights to its members, aiming for a more scalable, reliable, and context-aware evaluation system.

**Strengths:**

1. The paper is well-written and clearly presents the core concepts of the proposed framework, making it easy for readers to understand the main ideas and contributions.

2. The authors validate their approach across a wide range of datasets for both summarization and RAG tasks. This extensive evaluation, despite potential issues in the experimental setup noted below, provides evidence for the method's generalizability.

**Weaknesses:**

1. **Systematic Evaluation of Judge Pool Composition**: The experiments are conducted using a fixed pool of 9 LLMs, which includes a mix of open and closed-source models of varying sizes. *This experimental choice could significantly influence the results**. It is unclear how the framework would perform with a different composition of judges, particularly a pool of models with **more comparable capabilities**. As shown in Figure 3 (Summarization Groundedness), the selection process appears to converge on a few specific models. A more comparable judge pool might lead to different outcomes and would also provide a more challenging test for the "**Static Jury**" baseline, which is naturally **disadvantaged by the inclusion of weaker LLMs**. An analysis of the framework's performance under different judge pool configurations is needed.

2. Potentially Flawed Human-LLM Alignment Metric: The paper's method for measuring alignment between an LLM judge and human experts is based on the *absolute difference* between their normalized scores. As seen in Figure 3, this leads to counter-intuitive results where a model like Claude 3.7 Sonnet is the top choice for "Summarization Groundedness" but is almost never selected for "Summarization Completeness". Rather than concluding that the model has poor alignment on completeness, it is more plausible that this is due to a **systematic scoring bias** (e.g., the model consistently gives higher scores than humans). Even with such a bias, the model's preference ranking of outputs might still align well with human judgment, a nuance that the current metric fails to capture (as suggested by the **rank correlations in Figure 2**, Claude 3.7 Sonnet still can achieve very high alignment with humans on "Summarization Completeness"). **Since this alignment score is the training label for the reliability predictors, any bias introduced by this metric is critical and warrants further investigation.**

3. **Insufficient Baseline Comparisons**: The conclusion that an ensemble of LLM judges outperforms a **single judge** is largely expected. To demonstrate the effectiveness, comparsion should be done with other ensemble methods. However, the authors only compare their dynamic jury to a simple averaging baseline (also with some weak LLMs in the pool). This is **insufficient to demonstrate the novelty and effectiveness** of the proposed selection mechanism. More advanced and relevant baselines should be included, such as:
    - A static "Top-K" jury, where the best-performing K models on the entire validation set are selected.
    - Methods from the LLM routing or mixture-of-experts literature, which are conceptually similar to the task of selecting the best model for a given instance.

**Questions:**

- Could you please clarify how the *Kendall's Tau correlation* was computed in your experiments? Specifically, does it measure the correlation of rankings over the entire test set for each task, or is it calculated differently?

---

> ### Author Response · Authors · 2025-11-21
> **Authors' Reply to Reviewer Comments-4.1**
>
> We thank the reviewer for their time and insightful comments on our
> submission, which highlighted important areas for clarification and
> future exploration.
>
> > **1\.Systematic Evaluation of Judge Pool Composition**: The experiments
> are conducted using a fixed pool of 9 LLMs, which includes a mix of open
> and closed-source models of varying sizes. *This experimental choice
> could significantly influence the results*\*. It is unclear how the
> framework would perform with a different composition of judges,
> particularly a pool of models with **more comparable capabilities**. As
> shown in Figure 3 (Summarization Groundedness), the selection process
> appears to converge on a few specific models. A more comparable judge
> pool might lead to different outcomes and would also provide a more
> challenging test for the \"**Static Jury**\" baseline, which is
> naturally **disadvantaged by the inclusion of weaker LLMs**. An analysis
> of the framework\'s performance under different judge pool
> configurations is needed.
>
> We agree it is important to check if
> including weaker models in the pool hurts performance and to understand
> the trade-off between judge strength and diversity. To test this, we
> first ran an experiment where we explicitly removed the weaker models, such as Phi-4-reasoning, Gemma 3-27B-IT, and LLAMA-3.2-3B-Instruct, and kept
> only a \"Strong Pool\" containing Claude 3.7, Gemini 2.5 Flash, Gemini
> 2.5 Pro, GPT-OSS-20B, and GPT-OSS-120B. We compared this smaller pool
> against the full 9-judge pool on three task-metric combinations:
> Summarization-Completeness, Summarization-Groundedness, and
> RAG-Groundedness.
>
> The results show that using the full 9-judge pool works better than or
> equal to the restricted strong-only pool. For Summarization-Completeness, the full pool achieved a mean Kendall's Tau of 0.51,
> while the strong-only pool dropped to 0.46. For
> Summarization-Groundedness, both pools achieved 0.55. For
> RAG-Groundedness, the full pool reached 0.68, outperforming the
> strong-only pool at 0.65. This demonstrates that our Jury-on-Demand
> framework correctly manages weaker models, using them when they add
> value and ignoring them when they don\'t, so removing them actually
> reduces the system\'s flexibility.
>
> We also conducted ablation experiments to analyze judge diversity. We
> defined diversity in two ways: metric-based (using pairwise Spearman
> correlation) and family-based (grouping by model architecture, such as
> Gemini family or GPT family). To strictly study the effect of pool
> composition, we modified our setup for this specific experiment: instead
> of using our dynamic selection (tuning \$K\$), we forced the system to
> use a fixed jury where the jury size \$K\$ equaled the total pool size.
> This ensures all models in a specific pool contributed to the score.
>
> We found that individual judge strength is consistently the dominant
> factor in this fixed-jury setting. The strongest judges, such as
> GPT-OSS-120B and Gemini 2.5 Flash, naturally show high pairwise
> correlation because they all converge on the ground truth (human
> annotations). Therefore, the \"Most Similar\" pool was effectively just
> the \"Strongest\" pool. Conversely, trying to force a \"Diverse\" pool
> required us to include weaker models to lower the correlation. The drop
> in strength from adding these weak models hurt performance more than the
> diversity helped.
>
> Our hypothesis is that a pool that is both highly strong *and* highly
> diverse would be ideal, but this is difficult to construct with current
> models because the strong ones are highly correlated. Crucially, these
> findings confirm the robustness of our approach. Users do not need to
> manually curate a perfectly balanced pool. If a pool contains highly
> correlated strong judges, JOD leverages their combined strength; if weak
> judges are added for the sake of diversity, our reliability predictors
> correctly identify their lower reliability and down-weight them.

---

> ### Author Response · Authors · 2025-11-21
> **Authors' Reply to Reviewer Comments-4.2**
>
> > **2\.Potentially Flawed Human-LLM Alignment Metric**: The paper\'s method for
> measuring alignment between an LLM judge and human experts is based on
> the *absolute difference* between their normalized scores. As seen in
> Figure 3, this leads to counter-intuitive results where a model like
> Claude 3.7 Sonnet is the top choice for \"Summarization Groundedness\"
> but is almost never selected for \"Summarization Completeness\". Rather
> than concluding that the model has poor alignment on completeness, it is
> more plausible that this is due to a **systematic scoring bias** (e.g.,
> the model consistently gives higher scores than humans). Even with such
> a bias, the model\'s preference ranking of outputs might still align
> well with human judgment, a nuance that the current metric fails to
> capture (as suggested by the **rank correlations in Figure 2**, Claude
> 3.7 Sonnet still can achieve very high alignment with humans on
> \"Summarization Completeness\"). **Since this alignment score is the
> training label for the reliability predictors, any bias introduced by
> this metric is critical and warrants further investigation.**
>
> Regarding your observation that Claude 3.7 Sonnet
> is frequently chosen for Summarization Groundedness but rarely for
> Summarization Completeness, we first note that Table 8 shows Claude 3.7
> Sonnet ranks fifth among single judges for Summarization Completeness,
> while it ranks first for Summarization Groundedness. Since
> the optimal jury size for Summarization Completeness is smaller (three
> judges), it is expected that Claude 3.7 Sonnet is not selected as
> often.
>
> We further examined the Pearson correlation between judges' scores for
> Summarization Completeness and found that Claude 3.7 Sonnet's scores are
> highly correlated with those of Gemini 2.5 Flash and Gemini 2.5 Pro
> (correlation around 0.8). Our results indicate that Gemini 2.5 Flash
> is frequently selected, while Claude 3.7 Sonnet and Gemini 2.5 Pro are
> rarely chosen because Gemini 2.5 Flash sufficiently represents the score
> distribution of all three judges.
>
> Finally, we note that some judges are selected more often despite weaker
> overall performance. This occurs because, although their global
> performance may be lower, they exhibit strong performance within
> specific datasets or regions of the data, patterns our model learns
> from text properties. This adaptive selection mechanism is one of the
> key novelties and main contributions of our work.
>
> Regarding your comments on scoring bias, we greatly appreciate this
> insightful observation, and it represents an important future direction
> for our work. In this submission, we partially address potential scoring
> bias by introducing a non-zero tolerance between human annotations and
> judge scores when defining labels for the XGBoost model, and by tuning
> this tolerance when constructing the jury.
>
> Additionally, to further mitigate scoring bias, we explored score
> calibration and included the corresponding analysis in Appendix J.
> Specifically, we performed isotonic calibration for each judge's score
> within each dataset, then retrained the XGBoost model and constructed
> the jury using the calibrated scores. For illustration, we focused
> on summarization completeness and groundedness. Prior to calibration, we
> examined the difference between each judge's mean score and the mean
> human annotation score (annotation score minus judge score).
> We observed from Figure 11 that weaker models, particularly LLAMA 3.2
> 3B Instruct, Gemma 3-27B-IT, and Phi 4 Reasoning, tend
> to exhibit larger discrepancies. Additionally, nearly all judges assign
> lower scores than human annotations for groundedness in Unisumm data,
> suggesting a potential need for calibration.
>
> We then examined the results after calibration. From Table 29 and 30, we
> found that overall performance changes are minimal, with calibrated
> results slightly improving on groundedness (0.63 vs. 0.612) but
> slightly worsening on completeness (0.476 vs. 0.49). Differences become
> more pronounced within certain datasets and for specific judges. For
> example, tau for Gemini 2.5 Flash on Unisumm completeness drops from
> 0.592 to 0.53 because many original score 5s are calibrated to 4,
> leading to underestimation of human annotation scores. Conversely,
> for Unisumm groundedness, Gemini 2.5 Flash's tau increases from 0.6 to
> 0.65 after calibration because scores of 2 and 3 are calibrated to 4,
> reducing bias in judge scores. In summary, calibration can be beneficial
> for certain tasks and judges, but it may also introduce under- or
> overestimation of human annotations, reducing alignment. We will explore
> more effective ways to mitigate judge score bias in future works.

---

> ### Author Response · Authors · 2025-11-21
> **Authors' Reply to Reviewer Comments-4.3**
>
> > **3\.Insufficient Baseline Comparisons**
>
> We agree that comparing only against a
> simple average is not enough to show the novelty of our dynamic
> selection mechanism compared to other ensemble methods. To fix this, we
> added the specific baselines requested to our evaluation in Section 4.2
> and Section 5.1.
>
> To address the request for a static \"Top-K\" jury, we
> implemented **Static Jury (Average-Top-K)**. This method picks the \$K\$
> best-performing judges based on their Kendall's Tau on the validation
> set and averages their scores. We also tuned the jury size \$K\$ using
> the validation data. To address the comment about routing and
> mixture-of-experts, we implemented **Static Jury
> (Weighted-Regression)**. This baseline trains a linear regression model
> without an intercept on the training set. It uses judge scores as
> features to predict human annotations, and the learned coefficients
> serve as the static weights. Additionally, we included **Static Jury
> (Weighted-Tau)** to compare against performance-weighted averaging. This
> uses the softmax-normalized Kendall's Tau of each judge on the
> validation set as static weights.
>
> Our updated results in Table 1 and Figure 2 show that
> the **Jury-on-Demand (JOD)** framework consistently beats these
> optimized baselines. In the overall RAG-Groundedness task, which
> aggregates CAQA, HaluEval, and RagTruth, JOD achieves a mean Kendall's
> Tau of 0.68. This is higher than Weighted-Regression (0.64), Static
> Jury Average-All (0.61), Average-Top-K (0.59), and Weighted-Tau (0.52).
> The gap is even bigger on difficult datasets like *RagTruth*. On this
> benchmark, JOD reaches a Kendall's Tau of 0.58, which significantly
> outperforms Average-Top-K (0.34) and Weighted-Tau (0.18), and also
> exceeds Weighted-Regression (0.55) and Average-All (0.54). These
> findings confirm that dynamically selecting and weighting judges for
> each instance works better than static methods, especially when static
> strategies like Top-K fail to generalize to hard subsets. (More details see table below)
>
> These results highlight that the core assumption of the
> Static Top-K baseline, that the judges who perform best *on
> average* are the best for *every* query, is often flawed. A judge that
> ranks 4th on average might be the single best expert for a specific,
> niche instance. By dynamically assembling the jury for each specific
> input, Jury-on-Demand acts as an effective instance-level
> Mixture-of-Experts router, capturing specific details that global static
> selection misses. We have updated the manuscript to include these
> detailed comparisons and analysis.
>
> ### Table: Summarization Completeness — Jury-on-Demand vs Static Jury Baselines
> *Numbers in parentheses are standard deviation.*
>
> | **Data**       | **Jury-on-Demand** | **Static Jury (Average-All)** | **Static Jury (Average-TopK)** | **Static Jury (Weighted-Regression)** | **Static Jury (Weighted-Tau)** |
> |-----------------|---------------------|--------------------------------|---------------------------------|----------------------------------------|---------------------------------|
> | **Overall**    | 0.51 (0.03)        | 0.46 (0.02)                   | 0.49 (0.03)                    | 0.48 (0.02)                           | 0.47 (0.02)                    |
> | **SummEval**   | 0.74 (0.05)        | 0.62 (0.05)                   | 0.68 (0.07)                    | 0.68 (0.06)                           | 0.61 (0.04)                    |
> | **TL;DR**      | 0.39 (0.06)        | 0.39 (0.03)                   | 0.45 (0.04)                    | 0.43 (0.03)                           | 0.40 (0.03)                    |
> | **UniSumEval** | 0.68 (0.05)        | 0.61 (0.05)                   | 0.63 (0.05)                    | 0.58 (0.05)                           | 0.62 (0.05)                    |
>
> ### Table: RAG Groundedness — Jury-on-Demand vs Static Jury Baselines
> *Numbers in parentheses are standard deviation.*
>
> | **Data**      | **Jury-on-Demand** | **Static Jury (Average-All)** | **Static Jury (Average-TopK)** | **Static Jury (Weighted-Regression)** | **Static Jury (Weighted-Tau)** |
> |---------------|---------------------|--------------------------------|---------------------------------|----------------------------------------|---------------------------------|
> | **Overall**   | 0.68 (0.02)        | 0.61 (0.02)                   | 0.59 (0.02)                    | 0.64 (0.02)                           | 0.52 (0.02)                    |
> | **CAQA**      | 0.67 (0.03)        | 0.62 (0.02)                   | 0.60 (0.03)                    | 0.62 (0.02)                           | 0.58 (0.03)                    |
> | **HalEval**   | 0.77 (0.02)        | 0.73 (0.02)                   | 0.74 (0.03)                    | 0.73 (0.02)                           | 0.57 (0.04)                    |
> | **RagTruth**  | 0.58 (0.04)        | 0.54 (0.05)                   | 0.34 (0.06)                    | 0.55 (0.02)                           | 0.18 (0.05)                    |

---

> ### Author Response · Authors · 2025-11-21
> **Authors' Reply to Reviewer Comments-4.4**
>
> > **Questions:**
> > Could you please clarify how the Kendall\'s Tau correlation was computed
> in your experiments? Specifically, does it measure the correlation of
> rankings over the entire test set for each task, or is it calculated
> differently?
>
> The Kendall's tau values were computed using
> the scipy.stats.kendalltau package. For the overall Tau reported in
> Table 8-11, the metric measures the correlation of rankings across the
> entire test set for each task. Specifically, we concatenate all datasets
> within a task and then compute Kendall's tau on this combined data.
>
> If there are still unresolved issues, please let us know any additional questions or suggestions that could help us further improve the submission.

---

### Official Review · Reviewer_GbAd · 2025-10-27

**Soundness:** 2
**Presentation:** 1
**Contribution:** 2
**Rating:** 2
**Confidence:** 4

**Summary:**

This paper proposes LLM Jury-on-Demand, a dynamic, learning-based framework for automated LLM evaluation that predicts instance-level reliability for each candidate judge and then assembles a context-aware jury whose members’ raw scores are weighted by predicted reliabilities. The system (1) extracts rich textual features from inputs/outputs (token/size/embedding/text-complexity features; Fig.1, p.3), (2) trains per-judge XGBoost classifiers to predict whether a judge’s score will be “good” vs “bad” relative to human scores (Sec.3.3), and (3) for each instance selects the top-K judges by predicted reliability and aggregates their scores via reliability-weighted averaging (Sec.3.4). Experiments on summarization and RAG metrics (groundedness, relevance, completeness) across multiple datasets show the Jury-on-Demand consistently improves Kendall’s Tau correlation with human judgments versus single-judge and static-ensemble baselines (Fig.2, Table 1). The paper analyzes judge selection patterns and feature importance (e.g., embedding PCA vs compression ratio) and provides ablations and implementation details in the appendix.

**Strengths:**

* Clear problem with practical solution. The paper addresses a concrete weakness of current LLM-as-judge approaches (judge reliability varies by instance) and proposes an implementable remedy (per-judge reliability predictors + dynamic jury selection).

* Practical gains. Jury-on-Demand consistently achieves higher Kendall’s Tau than baselines; improvements are shown at both aggregate and dataset levels

**Weaknesses:**

Several areas limit the paper’s completeness and broader impact:

* First, the method’s sensitivity to hyperparameters (τ for reliability labeling, K for jury size) is not systematically analyzed; these choices likely influence both reliability prediction and final performance.
* Second, baselines could be stronger, comparing against static weighted ensembles or calibration-based weighting would clarify the added value of instance-level adaptivity.
* Third, statistical significance and effect sizes are not explicitly reported, even though claims of superiority rely on small Tau differences.

**Questions:**

N/A

---

> ### Author Response · Authors · 2025-11-21
> **Authors' Reply to Reviewer Comments-3.1**
>
> We thank the reviewer for their time and insightful comments on our
> submission, which highlighted important areas for clarification and
> future exploration.
>
> > 1\.First, the method's sensitivity to hyperparameters ($\tau$ for reliability
> labeling, $K$ for jury size) is not systematically analyzed; these choices
> likely influence both reliability prediction and final performance.
>
> To address this concern, we have conducted a detailed analysis, which is now presented in Appendix H.2 and H.3.
>
> - **Analysis on $K$**
>
> We focus on the tasks Summarization-Completeness and RAG-Groundedness
> for this study. The tolerance level for all trained XGBoost models is set to 0 which means
> only the exactly matching scores are considered as correct. Performance is measured across
> 10 runs and the average is taken for each jury size. Fig. 10 in the paper and the table below show that the performance varies with jury size. In the Summarization-Completeness task it gives the best performance with a jury size of 3. Performance first increases with increasing jury size as better judges together provide a better overall decision. It then decreases as poor judges need to be added to the jury, impacting the overall jury decision. In the RAG-Groundedness task performance is poor at lower jury sizes and improves with bigger jury sizes. In this case, adding more judges benefits the overall jury decision by making it more robust against noisy predictions. This shows us that tuning on jury size gives improvements than keeping a fixed one from task to task.
>
> *Table: Kendall's tau for jury performance with different number of judges in the jury.*
>
> | **K**            | 1     | 2     | 3     | 4     | 5     | 6     | 7     | 8     | 9     |
> |-------------------|-------|-------|-------|-------|-------|-------|-------|-------|-------|
> | **Summ Completeness** | 0.494 | 0.499 | 0.504 | 0.499 | 0.499 | 0.491 | 0.485 | 0.478 | 0.463 |
> | **RAG Groundedness**    | 0.643 | 0.664 | 0.670 | 0.682 | 0.683 | 0.682 | 0.677 | 0.681 | 0.681 |
>
> - **Analysis on Tolerance $\tau$**
>
> For this study, we focus on Summarization-Completeness. We experiment with two tolerance
> levels: 0 and 1 in the original scale (1-5). After min-max normalization to $[0, 1]$, these
> correspond to 0 and 0.25. The performance of the jury is observed with all the XGBoost
> models trained either on tolerance of 0 or 0.25. The table below (Table 25 in the paper) summarizes the means of the Kendall’s Tau across the 10 runs and shows the comparison with
> tuned tolerance models. We observe that allowing different tolerance levels across different
> XGBoost models gives slightly better performance than a fixed tolerance level across all
> models.
>
> Additionally, The table below (Table 25 in the paper) further illustrates the importance of tolerance tuning. While Jury-on-Demand with variable tolerance achieves the best overall performance, the optimal fixed tolerance differs across datasets: TL;DR performs best with a tolerance of 0, whereas UniSumEval favors 0.25. This variability underscores that no single fixed tolerance can fit all datasets. In practical scenarios, especially for unseen datasets without human annotations, it is impossible to know the ideal tolerance beforehand. Therefore, adaptive
> approaches that allow tolerance to vary across models or instances are crucial for robust
> generalization.
>
> Finally, the results with fixed tolerance levels are better than the static jury as we have
> chosen the best jury size ($K = 3$) overall across the runs. These findings suggest that while
> hyperparameter tuning enhances performance, the method remains reasonably robust under alternative settings.
>
> *Table: Kendall’s tau for jury performance on Summarization-Completeness under tolerance
> ablation. The number of judges (K) is fixed to 3 (overall best) for the fixed tolerance
> configurations.*
>
> | **Data**      | **Fixed tolerance (0)** | **Fixed tolerance (0.25)** | **Jury on demand (variance)** | **Static Jury** |
> |---------------|--------------------------|-----------------------------|--------------------------------|------------------|
> | **Overall**   | 0.49            | 0.50               | 0.51                   | 0.46    |
> | **SummEval**  | 0.70             | 0.70               | 0.74                   | 0.62     |
> | **TL;DR**     | 0.46           | 0.50               | 0.39                 | 0.39     |
> | **UniSumEval**| 0.71            | 0.66              | 0.68                   | 0.61    |

---

> ### Author Response · Authors · 2025-11-21
> **Authors' Reply to Reviewer Comments-3.2**
>
> > 2\.Second, baselines could be stronger, comparing against static weighted
> ensembles or calibration-based weighting would clarify the added value
> of instance-level adaptivity.
>
> We agree that relying solely on a simple
> unweighted average does not fully capture the efficacy of our
> instance-level approach relative to stronger static ensembles.
> Consequently, we have expanded our evaluation in Section 4.2 and Section
> 5.1 to include three rigorous static baselines: a performance-weighted
> ensemble, a regression-based router, and an optimized subset average.
>
> Specifically, we implemented a Static Jury (Weighted-Tau), which uses
> the softmax-normalized Kendall's Tau of each judge on the validation set
> as static weights. We also added a Static Jury (Weighted-Regression),
> which is a linear regression model trained without an intercept on the
> training set; this model uses judge scores as features to predict human
> scores, with the learned coefficients serving as the aggregation
> weights. Finally, we introduced a Static Jury (Average-Top-K) baseline
> that averages the scores of the K best-performing judges, where
> performance is measured by Kendall's Tau on the validation set and
> the optimal size K is also tuned on validation data.
>
> Our updated results, presented in Table 1 and Figure 2, demonstrate that
> the Jury-on-Demand (JOD) framework consistently outperforms these
> optimized baselines. In the overall RAG-Groundedness task---aggregated
> across the CAQA, HaluEval, and RagTruth datasets---JOD achieves a mean
> Kendall's Tau of 0.68, surpassing the Weighted-Regression (0.64), Static
> Jury Average-All (0.61), Average-Top-K (0.59), and Weighted-Tau (0.52)
> baselines. This performance gap becomes even more pronounced on
> specific, challenging datasets such as RagTruth. On this benchmark, JOD
> achieves a Kendall's Tau of 0.58, significantly outperforming the
> Average-Top-K (0.34) and Weighted-Tau (0.18) baselines, while also
> exceeding the Weighted-Regression (0.55) and Average-All (0.54) models.
> These findings confirm that dynamically selecting and weighting judges
> at the instance level provides a robustness that even optimized static
> ensembles cannot match, particularly when static selection methods
> like Top-K fail to generalize to difficult subsets. We have updated the
> manuscript to include these detailed comparisons. (For more details see the table below.)
>
> ### Table: Summarization Completeness — Jury-on-Demand vs Static Jury Baselines
> *Numbers in parentheses are standard deviation.*
>
> | **Data**       | **Jury-on-Demand** | **Static Jury (Average-All)** | **Static Jury (Average-TopK)** | **Static Jury (Weighted-Regression)** | **Static Jury (Weighted-Tau)** |
> |-----------------|---------------------|--------------------------------|---------------------------------|----------------------------------------|---------------------------------|
> | **Overall**    | 0.51 (0.03)        | 0.46 (0.02)                   | 0.49 (0.03)                    | 0.48 (0.02)                           | 0.47 (0.02)                    |
> | **SummEval**   | 0.74 (0.05)        | 0.62 (0.05)                   | 0.68 (0.07)                    | 0.68 (0.06)                           | 0.61 (0.04)                    |
> | **TL;DR**      | 0.39 (0.06)        | 0.39 (0.03)                   | 0.45 (0.04)                    | 0.43 (0.03)                           | 0.40 (0.03)                    |
> | **UniSumEval** | 0.68 (0.05)        | 0.61 (0.05)                   | 0.63 (0.05)                    | 0.58 (0.05)                           | 0.62 (0.05)                    |
>
> ### Table: RAG Groundedness — Jury-on-Demand vs Static Jury Baselines
> *Numbers in parentheses are standard deviation.*
>
> | **Data**      | **Jury-on-Demand** | **Static Jury (Average-All)** | **Static Jury (Average-TopK)** | **Static Jury (Weighted-Regression)** | **Static Jury (Weighted-Tau)** |
> |---------------|---------------------|--------------------------------|---------------------------------|----------------------------------------|---------------------------------|
> | **Overall**   | 0.68 (0.02)        | 0.61 (0.02)                   | 0.59 (0.02)                    | 0.64 (0.02)                           | 0.52 (0.02)                    |
> | **CAQA**      | 0.67 (0.03)        | 0.62 (0.02)                   | 0.60 (0.03)                    | 0.62 (0.02)                           | 0.58 (0.03)                    |
> | **HalEval**   | 0.77 (0.02)        | 0.73 (0.02)                   | 0.74 (0.03)                    | 0.73 (0.02)                           | 0.57 (0.04)                    |
> | **RagTruth**  | 0.58 (0.04)        | 0.54 (0.05)                   | 0.34 (0.06)                    | 0.55 (0.02)                           | 0.18 (0.05)                    |

---

> ### Author Response · Authors · 2025-11-21
> **Authors' Reply to Reviewer Comments-3.3**
>
> > 3\.Third, statistical significance and effect sizes are not explicitly
> reported, even though claims of superiority rely on small Tau
> differences.
>
> To address this, we added the statistical
> significance and effect sizes in Appendix E.1 Table 12 to 15.
> Specifically, we performed one-sided Wilcoxon signed-rank tests to
> compare the Tau differences between Jury-on-Demand and either static
> juries or single judges and reported the p-values. Among these p-values,
> 82% are statistically significant (p \< 0.05). We also computed Cliff's
> delta, a non-parametric effect size metric that quantifies the
> difference between two groups, with Jury-on-Demand as the baseline.
> According to conventional thresholds, an effect size is considered large
> if it exceeds 0.47 and medium if it falls between 0.33 and 0.47. Across
> all Cliff's delta values, 82% are classified as either large (72%) or
> medium (10%). This high proportion of significant p-values and
> substantial effect sizes indicates that, in most cases, Jury-on-Demand
> outperforms static baselines and single judges.
>
> If there are still unresolved issues, please let us know any additional questions or suggestions that could help us further improve the submission.

---

> > ### Comment · Reviewer_GbAd · 2025-11-21
> > **Thanks for the response**
> >
> > The authors addressed the missing experiment I proposed in the review, so I decided to increase my score for this paper. My further suggestions for this work are:
> >
> > 1. It seems like the proposed Jury-on-Demand does not perform well when the number of LLM judges is scaling up. I recommend choosing 1. a strong baseline and 2. the vanilla baseline to conduct the hyperparameter analysis on K and compare the performance with Jury-on-Demand. So it could be clear how Jury-on-Demand performs in large Jury settings.
> >
> > 2. Add more discussion about the advantages and limitations of Jury-on-Demand compared with the added strong baselines.

---

> > > ### Author Response · Authors · 2025-11-25
> > > **Author's Reply to Reviewer GbAd-Part 1**
> > >
> > > We sincerely appreciate your prompt and constructive feedback, as well as your decision to increase our score. Your comments have been invaluable in improving our work. Below, we present additional analyses based on your suggestions.
> > >
> > > > 1\.It seems like the proposed Jury-on-Demand does not perform well when the number of LLM judges is scaling up. I recommend choosing 1. a strong baseline and 2. the vanilla baseline to conduct the hyperparameter analysis on K and compare the performance with Jury-on-Demand. So it could be clear how Jury-on-Demand performs in large Jury settings.
> > >
> > > To further examine how performance varies with the number of judges $K$, we focused on two tasks for illustration: Summarization-Completeness and RAG-Groundedness. We compared Jury-on-Demand (JOD) with two static jury baselines:
> > >
> > > - Simple Average (Vanilla): The unweighted average of judge scores.
> > > - Weighted Regression: A linear regression model (without intercept) trained on the training set, using judge scores as features to predict human scores. The learned coefficients serve as aggregation weights.
> > >
> > > For these static jury baselines, we first determine which judges to include. Specifically, we rank judges by their Kendall’s tau performance for each task and select the top $K$ judges for the static jury. Table 1 shows the judge ranked by performance for each task. Tables 2 and 3 illustrate how Kendall’s tau changes with jury size $K$ for Summarization-Completeness and RAG-Groundedness, respectively. Results are averaged over 10 runs. Our main observations are:
> > >
> > > ### 1. JOD consistently outperforms static juries
> > > Performance varies with jury size $K$, but JOD consistently outperforms both static baselines across all $K$ values. The performance margin of JOD over static juries is larger for RAG-Groundedness than for Summarization-Completeness.
> > >
> > > ### 2. Effect of jury size differs by task
> > > - For Summarization-Completeness, all three methods perform worse as $K$ becomes large (e.g., $K = 7, 8, 9$), although Weighted Regression performs slightly better at $K = 9$.
> > > - For RAG-Groundedness, performance generally improves with larger $K$, except that Simple Average shows a drop at $K = 9$.
> > >
> > > To explain these trends, we examined the difference between the mean human annotation score and the mean score of weaker judges (Human – Judge), shown in Table 4. The magnitude of this bias is much larger for Summarization-Completeness than for RAG-Groundedness (except for Gemma-3-27B-IT). Large biases degrade performance when these judges are included, especially for Simple Average, which assigns equal weight regardless of bias. For RAG-Groundedness, most biases are moderate and can even improve performance through diversity. The drop for Simple Average at $K = 9$ occurs because Gemma-3-27B-IT (with the largest bias) is included.
> > >
> > > ### 3. Weighted Regression struggles with small K
> > > Weighted Regression performs worse than other jury methods when $K$ is small, particularly for RAG-Groundedness. One contributing factor is that the regression coefficients for RAG-Groundedness are very small, which results in weighted regression scores that fall below human annotation scores. To ensure realistic outputs, we round up any weighted regression score that is lower than the minimum human annotation score. However, because the coefficients are small, this adjustment often causes many weighted regression scores to equal the lowest human annotation score, further degrading performance. Another potential reason for poor performance at small $K$ is the high correlation among strong judges (e.g., Gemini models, GPT-OSS models, Claude 3.7 Sonnet), with correlation values around 0.7–0.8. Using such highly correlated features in regression can negatively impact model performance.
> > >
> > > In summary, all three methods exhibit consistent behavior within each task, with JOD always achieving the best performance. How performance changes with jury size $K$ depends on multiple factors, including task characteristics, score distributions, and correlations among judge scores.
> > >
> > > *Table 1: Judges ranked by Kendall’s tau (highest to lowest). Gemn. 2.5 is Gemini 2.5 Flash.*
> > >
> > > | **K**            | 1        | 2     | 3     | 4     | 5     | 6     | 7     | 8     | 9     |
> > > |-------------------|-----------|-------|-------|-------|-------|-------|-------|-------|-------|
> > > | **Summ Completeness** | GPT-OSS-120B | Gemn. 2.5 | Gemn. 2.0 | Gemn. 2.5 Pro | Claude 3.7 | GPT-OSS-20B | Gemma 3 | Phi 4 | Llama 3.2 |
> > > | **RAG Groundedness**  | GPT-OSS-120B | Gemn. 2.5 | GPT-OSS-20B | Gemn. 2.5 Pro | Claude 3.7 | Gemn. 2.0 | Phi 4 | Llama 3.2 | Gemma 3 |

---

> > > ### Author Response · Authors · 2025-11-25
> > > **Author's Reply to Reviewer GbAd-Part 4**
> > >
> > > ### **Limitations**
> > >
> > > 1. Larger Training Data
> > > Training JOD requires a substantial amount of data with high-quality human annotations. This poses challenges in sourcing large annotated datasets. Moreover, training and tuning JOD is more complex and time-consuming compared to static juries such as simple average or weighted regression.
> > >
> > > 2. Judge Score Bias
> > > As shown in Table 4 under previous reply, some weaker judges consistently assign higher or lower scores relative to human annotations. While we tune the tolerance parameter ($\tau$) to mitigate this issue, biased judge scores can still affect final jury performance.  We explore score calibration in Appendix J. Our experiments show that calibration sometimes improves alignment between judge scores and human annotations, but in other cases, it amplifies bias. Addressing this challenge remains an open problem, and we plan to investigate more robust approaches in future work.
> > >
> > > 3. Reliability Score Estimation
> > > For certain tasks, reliability score estimation may be less accurate. Tables 5 and 6 show that while weaker judges generally have lower reliability scores, the difference between strong and weak judges varies by task. For example, in RAG groundedness, the strong judge’s reliability score is much higher than the weaker ones, whereas in summarization completeness, the difference is less pronounced. Although reliability-based weighting is still more accurate than simple averaging, higher reliability scores for weaker judges can risk overweighting them, reducing jury accuracy. Improving reliability score prediction is another direction for future work.
> > >
> > >
> > > We will include these analyses in the camera-ready version. Thank you again for your helpful suggestions. Please let us know if you have any additional questions or recommendations that could help us further improve the submission.

---

> ### Author Response · Authors · 2025-11-25
> **Author's Reply to Reviewer GbAd-Part 2 (Tables)**
>
> *Table 2: Summarization Completeness: Kendall’s tau across varying jury sizes for different baselines.*
>
> | **K**            | 2     | 3     | 4     | 5     | 6     | 7     | 8     | 9     |
> |-------------------|-------|-------|-------|-------|-------|-------|-------|-------|
> | **Jury-on-Demand** | 0.499 | 0.504 | 0.499 | 0.499 | 0.491 | 0.485 | 0.478 | 0.463 |
> | **Static (Average K)**    | 0.458 | 0.472 | 0.483 | 0.478 | 0.470 | 0.449 | 0.441 | 0.454 |
> | **Static (Weighted-Regression)**    | 0.460 | 0.449 | 0.454 | 0.456 | 0.466 | 0.444 | 0.443 | 0.460 |
>
>
> *Table 3: RAG Groundedness: Kendall’s tau across varying jury sizes for different baselines.*
>
> | **K**            | 2     | 3     | 4     | 5     | 6     | 7     | 8     | 9     |
> |-------------------------------|-------|-------|-------|-------|-------|-------|-------|-------|
> | **Jury-on-Demand**         | 0.664 | 0.670 | 0.682 | 0.683 | 0.682 | 0.677 | 0.681 | 0.681 |
> | **Static (Average K)**     | 0.643 | 0.651 | 0.648 | 0.646 | 0.649| 0.650 | 0.644 | 0.614 |
> | **Static (Weighted-Regression)**  | 0.584 | 0.576 | 0.562 | 0.545 | 0.516 | 0.644 | 0.645 | 0.644 |
>
> *Table 4: Difference between average human annotation scores and average weaker judge scores (human − judge).*
>
> | **Judge**       | **LLAMA-3.2-3B-Instruct** | **Gemma-3-27B-IT** | **Phi-4-reasoning** |
> |-----------------|--------------------------|-----------------------------|-----------------------------|
> | **Summ Completeness**  | -0.40            | -0.37              | 1.64                   |
> | **RAG Groundedness**  | -0.21             | -0.75               | -0.09                   |

---

> ### Author Response · Authors · 2025-11-25
> **Author's Reply to Reviewer GbAd-Part 3**
>
> > 2\.Add more discussion about the advantages and limitations of Jury-on-Demand compared with the added strong baselines.
>
> Here are additional discussions on the main advantages and limitations of Jury-on-Demand (JOD) compared to the simple average and weighted regression jury baselines.
>
> ### **Advantages**
>
> 1. Dynamic Judge Selection
> Tables 8–11 in Appendix E.1 show that different judges perform differently across datasets, indicating that judge performance varies with dataset characteristics and text properties. Static juries, which use a fixed set of judges for all data, cannot capture these variations. In contrast, our proposed dynamic JOD selects the optimal judge set for each data instance, effectively adapting to these differences. Furthermore, JOD outperforms juries that use judge scores as features (e.g., weighted-regression jury), where correlated judge score features can degrade model performance.
>
> 2. Adaptive Judge Weights
> JOD assigns weights to judges based on reliability scores learned by the XGBoost model, reflecting each judge’s performance for individual instances. This ensures weaker models receive appropriately lower weights. For example, Tables 5 and 6 below report the average reliability scores for one strong judge (Gemini 2.5 Flash) and three weaker judges (LLAMA-3.2-3B-Instruct, Gemma-3-27B-IT, Phi-4-reasoning) on summarization completeness and RAG groundedness. Weaker judges consistently receive lower weights than the strong judge, preventing their scores from disproportionately influencing the final jury score.  In contrast, a static simple average jury assigns equal weights to all judges, which can overemphasize weaker models and reduce performance. Additionally, JOD guarantees weighted scores remain within a valid range, whereas regression-based juries may produce unrealistic values (e.g., exceeding the maximum human annotation score or falling below the minimum).
>
> *Table 5: Summarization Completeness: Average reliability scores.*
>
> | **Judge**       | **GEMINI_2_5_FLASH** | **LLAMA-3.2-3B-Instruct** | **Gemma-3-27B-IT** | **Phi-4-reasoning** |
> |-----------------|---------------------|--------------------------------|---------------------------------|----------------------------------------|
> | **Overall**    | 0.336        | 0.232                   | 0.284                    | 0.189                           |
> | **SummEval**   | 0.282        | 0.213                   | 0.305                    | 0.143                           |
> | **TL;DR**      | 0.308        | 0.198                  | 0.249                    | 0.204                           |
> | **UniSumEval** | 0.429        | 0.309                   | 0.327                    | 0.201                           |
>
>
> *Table 6: RAG Groundedness: Average reliability scores.*
>
> | **Data**       | **GEMINI_2_5_FLASH** | **LLAMA-3.2-3B-Instruct** | **Gemma-3-27B-IT** | **Phi-4-reasoning** |
> |-----------------|---------------------|--------------------------------|---------------------------------|----------------------------------------|
> | **Overall**    | 0.681        | 0.337                   | 0.427                    | 0.271                           |
> | **CAQA**   | 0.622        | 0.336                   | 0.338                    | 0.288                           |
> | **HaluEval**      | 0.796        | 0.262                  | 0.551                    | 0.126                           |
> | **RagTruth** | 0.629        | 0.407                   | 0.394                    | 0.389                           |

---

### Official Review · Reviewer_S8Cd · 2025-10-31

**Soundness:** 3
**Presentation:** 2
**Contribution:** 2
**Rating:** 2
**Confidence:** 2

**Summary:**

This paper addresses the limitations of existing Large Language Model (LLM) evaluation methods—slow and costly human evaluation, biased single LLM judges, and inflexible static juries—by proposing LLM Jury-on-Demand, a dynamic, learning-based evaluation framework. The framework first extracts multi-dimensional text features (text size, special words, complexity, and embedding-related features) from input data (e.g., source text, generated summaries for summarization tasks). It then trains XGBoost-based reliability predictors to assess how well each LLM judge aligns with human experts, dynamically selects the most reliable judges to form an optimal jury for each data point, and aggregates scores using reliability as weights.

**Strengths:**

* Introduces a creative combination of existing ideas (LLM-as-a-Judge, multi-model juries, LLM performance prediction) into the LLM Jury-on-Demand framework.

* The methodology is rigorously designed: (1) Multi-dimensional feature engineering covers key signals for reliability prediction; (2) XGBoost-based reliability predictors are tailored to each "judge-task-metric" combination, ensuring targeted prediction; (3) Experiments use 9 diverse LLMs as judges, span 2 core tasks (summarization/RAG) and multiple datasets.

* The work targets a critical pain point in high-stakes LLM deployment—lack of scalable, reliable evaluation.

**Weaknesses:**

* Over-Reliance on Human-Annotated Data Limits Scalability: Reliability predictors depend entirely on human-annotated datasets for training，this creates a critical bottleneck, high-stakes domains (e.g., legal/medical) often lack such annotations, making the framework hard to deploy there.

* Insufficient Analysis of Jury Size (K) and Tolerance (τ) Hyperparameters: K and τ are optimized via validation sets but provides no details on: (1) the search range of K (e.g., 3–9 judges) and how different K values impact performance; (2) why τ varies across judges/tasks or its sensitivity to results.

**Questions:**

NO ETHICS STATEMENT and REPRODUCIBILITY STATEMENT

Please see the weaknesses.

---

> ### Author Response · Authors · 2025-11-21
> **Authors' Reply to Reviewer Comments-2.1**
>
> We thank the reviewer for their time and insightful comments on our
> submission, which highlighted important areas for clarification and
> future exploration.
>
> > 1\.Over-Reliance on Human-Annotated Data Limits Scalability: Reliability
> predictors depend entirely on human-annotated datasets for training, this
> creates a critical bottleneck, high-stakes domains (e.g., legal/medical)
> often lack such annotations, making the framework hard to deploy there.
>
> We acknowledge that human-annotated data is
> needed to train the reliability models. As more and more annotated data
> becomes available, this will be of less concern in the future. In
> addition, some of the learned patterns can be generalized to unseen
> domains, which further mitigate the need for
> annotations. We added experiments in Appendix I to assess
> the framework's generalizability by training on a subset of domains and
> applying it to held-out domains in our data. In the RAG relevance
> task, across all three held-out data, jury-on-demand achieves the
> highest performance of 0.59, 0.89 and 0.46 comparing to static-jury or
> single judges (See table below - RAG Relevance), indicating that the trained
> framework generalizes effectively to unseen domains in this setting. In
> the Summarization Groundedness case, which includes four data
> sources, jury-on-demand demonstrates strong generalization for three of
> them, while performance on DialSummEval is weaker (See table below - Summarization Groundedness).
>
>
> *Table: Kendall's tau on held-out data source - RAG Relevance*
> | Held-out    | Jury-on-Demand | Static-Jury | Claude 3.7 | Gemn. 2.0 | Gemn. 2.5 | Gemn. 2.5 Pro | GPT-OSS-20B | GPT-OSS-120B | LL-3.2 | Gemma 3 | Phi-4 |
> |-------------|----------------|-------------|------------|-----------|-----------|---------------|-------------|--------------|--------|---------|-------|
> | **ALCE**        | 0.59           | 0.58        | 0.52       | 0.51      | 0.53      | 0.55          | 0.57        | 0.58         | 0.11   | 0.43    | 0.12  |
> | **HotpotQA**   | 0.89           | 0.87        | 0.85       | 0.84      | 0.88      | 0.88          | 0.88        | 0.88         | 0.40   | 0.77    | 0.46  |
> | **MS-MARCO**    | 0.46           | 0.42        | 0.42       | 0.39      | 0.43      | 0.43          | 0.43        | 0.43         | 0.14   | 0.17    | 0.12  |
>
> *Table: Kendall's tau on held-out data source - Summarization Groundedness*
> | Held-out        | Jury-on-Demand | Static-Jury | Claude 3.7 | Gemn. 2.0 | Gemn. 2.5 | Gemn. 2.5 Pro | GPT-OSS-20B | GPT-OSS-120B | LL-3.2 | Gemma 3 | Phi-4 |
> |-----------------|----------------|-------------|------------|-----------|-----------|---------------|-------------|--------------|--------|---------|-------|
> | **DialSummEval**  | 0.63           | 0.67        | 0.61       | 0.64      | 0.71      | 0.70          | 0.65        | 0.68         | 0.28   | 0.66    | 0.27  |
> | **SummEval**       | 0.64           | 0.62        | 0.60       | 0.55      | 0.62      | 0.62          | 0.54        | 0.60         | 0.20   | 0.57    | 0.14  |
> | **TL;DR**           | 0.43           | 0.43        | 0.36       | 0.40      | 0.38      | 0.43          | 0.27        | 0.40         | 0.08   | 0.38    | 0.07  |
> | **UniSummEval**   | 0.68           | 0.65        | 0.61       | 0.59      | 0.63      | 0.64          | 0.63        | 0.63         | 0.17   | 0.58    | 0.35  |

---

> ### Author Response · Authors · 2025-11-21
> **Authors' Reply to Reviewer Comments-2.2**
>
> > 2\.Insufficient Analysis of Jury Size ($K$) and Tolerance ($\tau$)
> Hyperparameters: $K$ and $\tau$ are optimized via validation sets but provides
> no details on: (1) the search range of $K$ (e.g., 3--9 judges) and how
> different $K$ values impact performance; (2) why $\tau$ varies across
> judges/tasks or its sensitivity to results.
>
> We have provided detailed analyses of the impact of $K$ and $\tau$, with results included in Appendix H.2 and H.3.
>
> - **Search Range and Impact of Jury Size $K$**
>
> The search range for $K$ spans from 1 to 9 (where $K = 1$ corresponds to selecting the best single judge). To examine the effect of jury size, we focus on the tasks Summarization Completeness and RAG Groundedness. The tolerance level for all trained XGBoost models is set to 0 which means only the exactly matching scores are considered as correct. Performance is measured across 10 runs and average is taken for each jury size. Fig. 10 in the paper and the table below show that the performance varies with jury size. In the Summarization-Completeness task it gives the best performance with a jury size of 3. Performance first increases with increasing jury size as better judges together provide a better overall decision. It then decreases as poor judges need to be added to the jury, impacting the overall jury decision. In the RAG-Groundedness task performance is poor at lower jury sizes and improves with bigger jury sizes. In this case, adding more judges benefits the overall jury decision by making it more robust against noisy predictions. This shows us that tuning on jury size gives improvements than keeping a fixed one from task to task.
>
> *Table: Kendall’s tau for jury performance with different number of judges in the jury.*
>
> | **K**            | 1     | 2     | 3     | 4     | 5     | 6     | 7     | 8     | 9     |
> |-------------------|-------|-------|-------|-------|-------|-------|-------|-------|-------|
> | **Summ Completeness** | 0.494 | 0.499 | 0.504 | 0.499 | 0.499 | 0.491 | 0.485 | 0.478 | 0.463 |
> | **RAG Groundedness**    | 0.643 | 0.664 | 0.670 | 0.682 | 0.683 | 0.682 | 0.677 | 0.681 | 0.681 |
>
> - **Tolerance ($\tau$) Variation and Sensitivity**
>
> Some judges may exhibit bias, such as consistently assigning higher or lower scores for certain data or tasks. This observation motivates our introduction of a tolerance parameter. For the sensitivity analysis on $\tau$, we focus on Summarization-Completeness. We experiment with two tolerance
> levels: 0 and 1 in the original scale (1-5). After min-max normalization to $[0, 1]$, these
> correspond to 0 and 0.25. The performance of the jury is observed with all the XGBoost
> models trained either on tolerance of 0 or 0.25. The table below (Table 25 in the paper) summarizes the means of the Kendall’s Tau across the 10 runs and shows the comparison with
> tuned tolerance models. We observe that allowing different tolerance levels across different
> XGBoost models gives slightly better performance than a fixed tolerance level across all
> models.
>
> Additionally, The table below (Table 25 in the paper) further illustrates the importance of tolerance tuning. While Jury-on-Demand with variable tolerance achieves the best overall performance, the optimal fixed tolerance differs across datasets: TL;DR performs best with a tolerance of 0, whereas UniSumEval favors 0.25. This variability underscores that no single fixed tolerance can fit all datasets. In practical scenarios, especially for unseen datasets without human annotations, it is impossible to know the ideal tolerance beforehand. Therefore, adaptive
> approaches that allow tolerance to vary across models or instances are crucial for robust
> generalization.
>
> Finally, the results with fixed tolerance levels are better than the static jury as we have
> chosen the best jury size ($K = 3$) overall across the runs. These findings suggest that while
> hyperparameter tuning enhances performance, the method remains reasonably robust under alternative settings.
>
> *Table: Kendall’s tau for jury performance on Summarization-Completeness under tolerance
> ablation. The number of judges (K) is fixed to 3 (overall best) for the fixed tolerance
> configurations.*
>
> | **Data**      | **Fixed tolerance (0)** | **Fixed tolerance (0.25)** | **Jury on demand (variance)** | **Static Jury** |
> |---------------|--------------------------|-----------------------------|--------------------------------|------------------|
> | **Overall**   | 0.49            | 0.50               | 0.51                   | 0.46    |
> | **SummEval**  | 0.70             | 0.70               | 0.74                   | 0.62     |
> | **TL;DR**     | 0.46           | 0.50               | 0.39                 | 0.39     |
> | **UniSumEval**| 0.71            | 0.66              | 0.68                   | 0.61    |
>
>
>
> If there are still unresolved issues, please let us know any additional questions or suggestions that could help us further improve the submission.

---

> ### Author Response · Authors · 2025-11-21
> **Authors' Reply to Reviewer Comments-2.3**
>
> > **Questions:**
> > NO ETHICS STATEMENT and REPRODUCIBILITY STATEMENT
>
> We have added the ethics statement and reproducibility statement at the end of the main text.

---

> > ### Comment · Reviewer_S8Cd · 2025-11-26
> > **comment**
> >
> > Thank you for your response. I have several remaining concerns:
> >
> > 1. What if we do not distinguish between task types—for example, using the same model for both summarization and RAG? In realistic scenarios, general-purpose settings are more common, and training separate models for each task type incurs significant overhead.
> >
> > 2. Given that you invoked multiple APIs, the cost seems disproportionate to the performance improvements achieved, raising questions about practical utility.
> >
> > 3. While the experiments and analysis are indeed comprehensive, the overall usability remains questionable.

---

> > > ### Author Response · Authors · 2025-12-01
> > > **Authors' Reply to Reviewer Comments-2.4**
> > >
> > > Thank you for your thoughtful review. Below, we provide clarifications addressing your concerns.
> > >
> > > > 1\.What if we do not distinguish between task types—for example, using the same model for both summarization and RAG? In realistic scenarios, general-purpose settings are more common, and training separate models for each task type incurs significant overhead.
> > >
> > > We understand that using a single model or even a single judge for evaluation across all tasks may seem straightforward. However, prior work ([1], [2]) and our experiments demonstrate the benefits of incorporating multiple judges that adapt to different tasks. Below are detailed explanations:
> > >
> > > ### 1. Different tasks have different feature distributions
> > > Summarization and RAG are fundamentally different tasks with distinct data structures: summarization involves a source text and a summary, while RAG includes a question, cited context, and an answer. Our models compute text features using all components of the input. These structural differences motivate the need for separate XGBoost reliability models tailored to each task. Experiments have demonstrated that judges have a various degrees of reliability. Figures 5–8 in Appendix E.2 show that jury composition differs across tasks and datasets, reflecting the variability in judge reliability across these contexts.
> > >
> > > ### 2. Different tasks require different model specialties
> > > Summarization primarily focuses on alignment between the summary and the source text, whereas RAG Q&A requires reasoning over both the question and the cited context. Consequently, a single model may perform inconsistently across tasks. Our experiments confirm this: Tables 9 and 11 in the paper show that GPT-OSS-120B achieves the best performance for summarization completeness, while Claude 3.7 Sonnet performs best for RAG completeness.
> > > Additionally, benchmarks such as HELM ([3]) indicate that top-performing models vary significantly across tasks, reinforcing the need for task-specific models.
> > >
> > > ### 3. Minimal overhead in practice
> > > After collecting judge scores, we train XGBoost models to compute reliability scores. XGBoost is highly efficient and computationally lightweight, so training separate models for different tasks does not introduce significant overhead. Moreover, training and tuning occur only once, not during inference.
> > >
> > > Finally, we emphasize that Jury-on-Demand improves generalization precisely because it is task-specific. Building juries that incorporate multiple models for different tasks is a key novelty of our framework and contributes to its superior performance.

---

> > > ### Author Response · Authors · 2025-12-01
> > > **Authors' Reply to Reviewer Comments-2.5**
> > >
> > > > 2\.Given that you invoked multiple APIs, the cost seems disproportionate to the performance improvements achieved, raising questions about practical utility.
> > >
> > > Regarding the concern about cost and practical utility, we do invoke multiple APIs, but this is a necessary step if we want to move beyond the limitations of a single judge. Whether using simple averaging or our method, a jury system inevitably requires multiple inputs. However, our framework specifically addresses this by tuning the parameter $K$. We work to find an optimal, smaller $K$ value to form a smaller jury, which allows us to invoke fewer APIs while maintaining high performance.
> > >
> > > If a user prefers not to call multiple APIs and stick to a single judge, our system is still highly valuable. **According to our experiments, no single judge performs consistently best across all cases.** For the decision of how to choose that single judge given a user’s inference data, our reliability score gives good guidance. It generates a more accurate jury score and provides guardrails. For example, even when used with a single judge, our reliability score can guide the user on whether the judge's score is reliable or if they should involve humans in the loop to make better decisions. Finally, for a tighter budget, one does not need to run all the judges. Users can choose to run only the top few judges, or even just the top one based on the reliability scores. This avoids the issue of using a fixed single judge that performs poorly in specific cases and provides the ensembling advantages of juries with a lower budget. Additionally, a key direction for our future work is to distill a model for the jury, which would significantly reduce the computational cost.
> > >
> > > Regarding the cost comparison, getting the check right is critical for high-stakes applications. The human cost for ensuring accuracy is still much more expensive than API calls. We can refer to **Gilardi et al. (2023)** ([4]), which states that "the per-annotation cost of ChatGPT is less than \$0.003---about thirty times cheaper than MTurk." MTurk is a crowd-worker platform, but in high-risk domains, the human annotation cost can be even higher because of the need for specialized training or domain experts. Since that paper was published in 2023, and it is now 2025, human costs likely have not decreased and may have increased, while API costs are getting much lower. Therefore, the cost difference is likely even greater than thirty times now. Additionally, **Jung et al. (2024)** ([5]) demonstrate that intelligent selection strategies can reduce evaluation costs by over 87\% compared to using only the strongest models like GPT-4, further proving that API-based evaluation is cost-effective.
> > >
> > > Finally, regarding the performance concern, the improvement varies case by case. For example, in Appendix E.1 of our paper, we see the improvement is quite decent for the RAG-Completeness task. For the overall dataset, Jury-on-Demand achieves a performance score of 0.49, compared to 0.36 for the best single judge. On the data of ASQA, Jury-on-Demand also scores 0.49, while the best single judge reaches only 0.38.

---

> > > ### Author Response · Authors · 2025-12-01
> > > **Authors' Reply to Reviewer Comments-2.6**
> > >
> > > > 3\.While the experiments and analysis are indeed comprehensive, the overall usability remains questionable.
> > >
> > > Regarding the usability of our framework, we designed it so that any user can focus almost entirely on the inference part. The reliability models and necessary hyperparameters (like tolerance levels) have already been trained and tuned. For the inference part, the user simply employs the pre-trained reliability models to generate reliability scores. They can use our tuned value $K$ as the jury size, or adapt their own smaller $K$ based on their specific budget. Then, for each instance, they run single judge scores only for the selected jury members and aggregate them to get a final jury score.
> > >
> > > As we mentioned in our previous response, we performed experiments regarding domain application which further supports usability. We found that our system generalizes well to held-out domains (See Appendix I). This means users can apply our system to new domains with high confidence without needing to retrain from scratch immediately.
> > >
> > > At the same time, the system is designed to be updated continuously. When we acquire new annotation data with high-quality human annotation scores, they will be added to the training data. Also, after the system has run for some time and we receive new inference data from users, we can update the model. We can choose to add inference instances where we have high confidence in our jury score back into the training set. For the low-confidence population, we can involve human annotation for those specific instances. All of this data will be added back to the training set, allowing the system reliability model to be retrained periodically to improve performance over time.
> > >
> > >
> > > We will incorporate these discussions into the camera-ready version. Thank you again for your valuable review and feedback, which have helped improve the quality of our work. Please let us know if these clarifications address your concerns.
> > >
> > >
> > > - References
> > >
> > > [1] Feng, S., Ding, W., Liu, A., Wang, Z., Shi, W., Wang, Y., Shen, Z., Han, X., Lang, H., Lee, C.-Y., et al. (2025). When One LLM Drools, Multi-LLM Collaboration Rules. arXiv preprint arXiv:2502.04506.
> > >
> > > [2] Verga, P., Hofstätter, S., Althammer, S., Su, Y., Piktus, A., Arkhangorodsky, A., Xu, M., White, N., & Lewis, P. (2024). Replacing Judges with Juries: Evaluating LLM Generations with a Panel of Diverse Models. arXiv preprint arXiv:2404.18796.
> > >
> > > [3] https://crfm.stanford.edu/helm/classic/latest/
> > >
> > > [4] Gilardi, F., Alizadeh, M., & Kubli, M. (2023). ChatGPT outperforms crowd-workers for text-annotation tasks. *Proceedings of the National Academy of Sciences*, 120(30), e2305016120.
> > >
> > > [5] Jung, J., Brahman, F., & Choi, Y. (2025). Trust or Escalate: LLM Judges with Provable Guarantees for Human Agreement. The Thirteenth International Conference on Learning Representations (ICLR).

---

### Official Review · Reviewer_byLJ · 2025-10-31

**Soundness:** 3
**Presentation:** 3
**Contribution:** 3
**Rating:** 6
**Confidence:** 4

**Summary:**

This work proposes a multi-LLM evaluation framework with dynamically selected judges for specific model contexts. The jury allocation mechanism is learned via a feature engineering technique from the instances of LLM inputs and outputs, where the task is to learn the most predictive features for judge reliabilities (alignment with human experts). With the learned scores for each LLM judge, the system is able to select the top performing judges and assemble their final evaluations with learned weights.  Experiments are conducted on a summarization benchmark and a RAG benchmark, where the results show the proposed dynamic system provide better alignment with manual evaluations than static and single jury baselines.

**Strengths:**

1 The learning-based design of feature selection is an effective way to discover judge reliabilities. Most of the candidate features are intuitive can be related to specific tasks or criteria.

2 The dynamic evaluation system is logically sound and technically proficient. One can easily identify the purpose of each component.

3 The experiment results show good alignment with human experts. The authors also give interesting and insightful findings on the analysis section.

4 The structure of this paper is well-organized and the writing is generally easy to understand.

**Weaknesses:**

1 This assembled jury method certainly brings more computation burden or api calls of multiple LLMs as juries, yet its performance does not seem to win by a large margin to justify the additional cost (e.g. on DialSumEval and SummEval, the best single Judge is only 0.01 lower than the proposed the multi-judge method)

2 The proposed method is also heavy on hyperparameter-tuning (K, multiple \tau(s), and multiple training parameters of XGboost). The paper should conduct ablation experiments on the robustness against more parameter settings, since it is unlikely to have a good validation set in real scenarios.

**Questions:**

please refer to weakness section.

---

> ### Author Response · Authors · 2025-11-21
> **Authors' Reply to Reviewer Comments-1.1**
>
> We thank the reviewer for their time and insightful comments on our
> submission, which highlighted important areas for clarification and
> future exploration.
>
> > 1\. This assembled jury method certainly brings more computation burden
> or api calls of multiple LLMs as juries, yet its performance does not
> seem to win by a large margin to justify the additional cost (e.g. on
> DialSumEval and SummEval, the best single Judge is only 0.01 lower than
> the proposed the multi-judge method)
>
> Regarding this concern, the improvement varies
> case by case. In Appendix E.1, we see the improvement is quite decent
> for the RAG completeness task. For the overall dataset,
> Jury-on-Demand achieves a performance score of 0.49, compared to 0.36
> for the best single judge. On the data of ASQA, Jury-on-Demand also
> scores 0.49, while the best single judge reaches 0.38. Secondly, our
> framework can not only generate a more accurate jury score,
> but also providing guard rails. For example, even used with a single
> judge, our reliability score can provide guidance on if the judge scores
> are reliable or not, hence involve human in the loop to make better
> decisions. Finally, one does not need to run all judges. For a tighter
> budget, one can run only the top few judges, or even top one judge based
> on the reliability scores. This avoids the issue of using a single
> fixed judge that doesn't do well for all cases. This provides
> the ensembling advantages of juries with a lower budget than a jury of
> fixed judges.  Additionally, a key direction for future work is to
> distill a model for the jury, which would significantly reduce
> computational cost.

---

> ### Author Response · Authors · 2025-11-21
> **Authors' Reply to Reviewer Comments-1.2**
>
> > 2\. The proposed method is also heavy on hyperparameter-tuning ($K$,
> multiple $\tau$(s), and multiple training parameters of XGboost). The
> paper should conduct ablation experiments on the robustness against more
> parameter settings, since it is unlikely to have a good validation set
> in real scenarios.
>
> We have conducted an extensive ablation study, now included in Appendix H.2 and H.3, to
> evaluate the robustness of our approach under varying parameter
> settings.
>
> - **Ablation Study on $K$**
>
> We focus on the tasks Summarization-Completeness and RAG-Groundedness
> for this study. The tolerance level for all trained XGBoost models is set to 0 which means
> only the exactly matching scores are considered as correct. Performance is measured across
> 10 runs and average is taken for each jury size. Fig. 10 in the paper and the table below show that the performance varies with jury size. In the Summarization-Completeness task it gives the best performance with a jury size of 3. Performance first increases with increasing jury size as better judges together provide a better overall decision. It then decreases as poor judges need to be added to the jury, impacting the overall jury decision. In the RAG-Groundedness task performance is poor at lower jury sizes and improves with bigger jury sizes. In this case, adding more judges benefits the overall jury decision by making it more robust against noisy predictions. This shows us that tuning on jury size gives improvements than keeping a fixed one from task to task.
>
> *Table: Kendall’s tau for jury performance with different number of judges in the jury.*
>
> | **K**            | 1     | 2     | 3     | 4     | 5     | 6     | 7     | 8     | 9     |
> |-------------------|-------|-------|-------|-------|-------|-------|-------|-------|-------|
> | **Summ Completeness** | 0.494 | 0.499 | 0.504 | 0.499 | 0.499 | 0.491 | 0.485 | 0.478 | 0.463 |
> | **RAG Groundedness**    | 0.643 | 0.664 | 0.670 | 0.682 | 0.683 | 0.682 | 0.677 | 0.681 | 0.681 |
>
> - **Ablation Study on Tolerance $\tau$**
>
> For this study, we focus on Summarization-Completeness. We experiment with two tolerance
> levels: 0 and 1 in the original scale (1-5). After min-max normalization to $[0, 1]$, these
> correspond to 0 and 0.25. The performance of the jury is observed with all the XGBoost
> models trained either on tolerance of 0 or 0.25. The table below (Table 25 in the paper) summarizes the means of the Kendall’s Tau across the 10 runs and shows the comparison with
> tuned tolerance models. We observe that allowing different tolerance levels across different
> XGBoost models gives slightly better performance than a fixed tolerance level across all
> models.
>
> Additionally, The table below (Table 25 in the paper) further illustrates the importance of tolerance tuning. While Jury-on-Demand with variable tolerance achieves the best overall performance, the optimal fixed tolerance differs across datasets: TL;DR performs best with a tolerance of 0, whereas UniSumEval favors 0.25. This variability underscores that no single fixed tolerance can fit all datasets. In practical scenarios, especially for unseen datasets without human annotations, it is impossible to know the ideal tolerance beforehand. Therefore, adaptive
> approaches that allow tolerance to vary across models or instances are crucial for robust
> generalization.
>
> Finally, the results with fixed tolerance levels are better than the static jury as we have
> chosen the best jury size ($K = 3$) overall across the runs. These findings suggest that while
> hyperparameter tuning enhances performance, the method remains reasonably robust under alternative settings.
>
> *Table: Kendall’s tau for jury performance on Summarization-Completeness under tolerance
> ablation. The number of judges (K) is fixed to 3 (overall best) for the fixed tolerance
> configurations.*
>
> | **Data**      | **Fixed tolerance (0)** | **Fixed tolerance (0.25)** | **Jury on demand (variance)** | **Static Jury** |
> |---------------|--------------------------|-----------------------------|--------------------------------|------------------|
> | **Overall**   | 0.49            | 0.50               | 0.51                   | 0.46    |
> | **SummEval**  | 0.70             | 0.70               | 0.74                   | 0.62     |
> | **TL;DR**     | 0.46           | 0.50               | 0.39                 | 0.39     |
> | **UniSumEval**| 0.71            | 0.66              | 0.68                   | 0.61    |
>
>
> If there are still unresolved issues, please let us know any additional questions or suggestions that could help us further improve the submission.

---

### Author Response · Authors · 2025-11-21
**Authors' Reply to Reviewer Comments - Global**

We sincerely thank all reviewers for their time, effort, and
constructive feedback, which has greatly helped us improve our work. To
address the reviewers' comments and provide a more comprehensive
analysis of our proposed jury approach, we conducted additional
experiments and updated the paper accordingly. Below is a summary of the
main revisions:

1.  **Stronger baselines:** We have expanded our evaluation in Section
    4.2 and Section 5.1 to include three rigorous static baselines: a
    performance-weighted ensemble, a regression-based router, and an
    optimized subset average. Specifically, we implemented a Static Jury
    (Weighted-Tau), which uses the softmax-normalized Kendall's Tau of
    each judge on the validation set as static weights. We also added a
    Static Jury (Weighted-Regression), which is a linear regression
    model trained without an intercept on the training set; this model
    uses judge scores as features to predict human scores, with the
    learned coefficients serving as the aggregation weights. Finally, we
    introduced a Static Jury (Average-Top-K) baseline that averages the
    scores of the K best-performing judges, where performance is
    measured by Kendall's Tau on the validation set and the optimal size
    K is also tuned on validation data. Our updated results, presented
    in Table 1 and Figure 2, demonstrate that the Jury-on-Demand (JOD)
    framework consistently outperforms these optimized baselines.

2.  **Analysis on parameters:** We have conducted an extensive ablation
    study, now included in Appendix H.2 and H.3, to evaluate the
    sensitivity and robustness of our approach under varying parameter
    settings. Specifically, we examined the impact of **jury size** and
    **tolerance levels** on performance. For jury size, we compared
    fixed versus tuned values on the summarization completeness and RAG
    groundedness tasks. Figure 10 shows that summarization-completeness
    peaks at K=3, while RAG-groundedness improves steadily with larger
    juries. This indicates that optimal jury size is task-dependent, and
    tuning K provides meaningful gains over fixed settings. For
    tolerance levels, we experimented with both fixed and mixed
    configurations across XGBoost models. Table 25 summarizes these
    results, showing that allowing different tolerance levels across
    models leads to improvements over a uniform setting.

3.  **Statistical significance and effect size:** We added the
    statistical significance and effect sizes in Appendix E.1 Table 12
    to 15. Specifically, we performed one-sided Wilcoxon signed-rank
    tests to compare the Tau differences between Jury-on-Demand and
    either static juries or single judges and reported the p-values.
    Among these p-values, 82% are statistically significant (p \< 0.05).
    We also computed Cliff's delta, a non-parametric effect size metric
    that quantifies the difference between two groups, with
    Jury-on-Demand as the baseline. Across all Cliff's delta values, 82%
    are classified as either large (72%) or medium (10%). This high
    proportion of significant p-values and substantial effect sizes
    indicates that, in most cases, Jury-on-Demand outperforms static
    baselines and single judges.

4.  **Score calibration:** To mitigate judge's scoring bias, we explored
    score calibration and included the corresponding analysis in
    Appendix J. Specifically, we performed isotonic calibration for each
    judge's score within each dataset, then retrained the XGBoost model
    and constructed the jury using the calibrated scores. For
    illustration, we focused on summarization completeness and
    groundedness. We found that overall performance changes are minimal,
    with calibrated results slightly improving on groundedness but
    slightly worsening on completeness. Differences become more
    pronounced within certain datasets for specific judges. In
    summary, calibration can be beneficial for certain tasks and judges,
    but it may also introduce under- or overestimation of human
    annotations, reducing alignment. We will explore more effective ways
    to mitigate judge score bias in future works.

5.  **Jury generalization:** We added experiments in Appendix I to
    assess the framework's generalizability by training on a subset of
    domains and applying it to held-out domains in our data. In the RAG
    relevance task, across all three held-out data, jury-on-demand
    achieves the highest performance of 0.59, 0.89, 0.46 comparing to
    static-jury or single judges (Table 27), indicating that the trained
    framework generalizes effectively to unseen domains.
    In the Summarization Groundedness case, which includes four data
    sources, jury-on-demand demonstrates strong generalization for three
    of them, while performance on the remaining one is weaker (Table
    28).

---

### Meta-Review · Area_Chair_myKw · 2025-12-06

**Summary:**

This paper introduces LLM Jury-on-Demand, a dynamic evaluation framework that improves automated LLM assessment. Instead of relying on a single judge or a fixed ensemble, the system predicts the instance-level reliability of multiple LLM judges using features extracted from model inputs and outputs (e.g., token length, embeddings, text complexity). Per-judge classifiers estimate whether a judge’s score will align with human experts, enabling the system to select the top-K most reliable judges and aggregate their evaluations with learned weights. Experiments on summarization and RAG datasets show consistently higher correlation with human judgments than static or single-judge baselines.

The key concern of this paper lies in hyperparameter selection. Although the authors provide additional ablation studies, the proposed Jury-on-Demand framework does not scale well as the number of LLM judges increases. Moreover, hyperparameter tuning is generally challenging, and it would be desirable to develop a more systematic method for identifying good hyperparameter configurations. Overall, while the paper has some merits, it requires major revisions. I encourage the authors to address the reviewers’ concerns and resubmit to a future venue.

**Reviewer Concerns:**

The key concerns raised by the reviewers involve the hyperparameter tuning of the proposed method. In the rebuttal, the authors provided additional ablation studies. However, the Jury-on-Demand framework still does not perform well when the number of LLM judges scales up.

**Reviewer Scores:**

Some reviewers agreed to raise their scores. However, since the original scores were low, the increases are unlikely to raise the overall rating above the acceptance threshold.

---

### Decision · Program_Chairs · 2026-01-26

Reject